# Cryo-tomography reveals rigid-body motion and organization of apicomplexan invasion machinery

Long Gui [1], William J. O'Shaughnessy [2], Kai Cai[1,4], Evan Reetz[1], Michael L. Reese [2,3] ✉ & Daniela Nicastro [1] ✉

The apical complex is a specialized collection of cytoskeletal and secretory machinery in apicomplexan parasites, which include the pathogens that cause malaria and toxoplasmosis. Its structure and mechanism of motion are poorly understood. We used cryo-FIB-milling and cryo-electron tomography to visualize the 3D-structure of the apical complex in its protruded and retracted states. Averages of conoid-fibers revealed their polarity and unusual nine-protofilament arrangement with associated proteins connecting and likely stabilizing the fibers. Neither the structure of the conoid-fibers nor the architecture of the spiral-shaped conoid complex change during protrusion or retraction. Thus, the conoid moves as a rigid body, and is not spring-like and compressible, as previously suggested. Instead, the apical-polar-rings (APR), previously considered rigid, dilate during conoid protrusion. We identified actin-like filaments connecting the conoid and APR during protrusion, suggesting a role during conoid movements. Furthermore, our data capture the parasites in the act of secretion during conoid protrusion.

All intracellular pathogens must accomplish entry into a new host cell. Apicomplexan parasites use an invasion machinery composed of a specialized cytoskeletal complex and secretory organelles. Collectively, this group of organelles is known as the apical complex. It is for this structure that the phylum Apicomplexa is named, which includes the causative agents of malaria, toxoplasmosis, and cryptosporidiosis. Apicomplexan parasites belong to the eukaryotic superphylum Alveolata, like ciliates and dinoflagellates (Supplementary Fig. 1). The apical complex is essential for parasite motility, and for invasion into and egress from host cells. Furthermore, mutations that disrupt the apical complex functions block the Apicomplexan lytic cycle, rendering them non-infectious[1–4].

The apical complex cytoskeleton itself is a -250 nm long structure (-25% the length of an *E. coli*) composed of a series of rings organized around a central spiral of specialized tubulin fibers called the conoid, and inside which secretory organelles are organized and prepared for secretion (Fig. 1a). Whereas the conoid fibers are composed of the same tubulin dimers that make up the parasites' subpellicular microtubules, they do not form closed tubes[5]. Instead, the conoid fibers form an open "C"-shaped structure[5]. The conoid complex is highly dynamic: it protrudes and retracts[6] as the parasites secrete adhesins and other motility/invasion factors[7,8]. Remarkably, the apical complex appears to have evolved from a eukaryotic cilium, as it contains both tubulin and cilium-associated proteins[9–12]. In addition, the core of the apical complex, including the conoid and its specialized tubulin structures, is conserved not only throughout Apicomplexa[13,14] and in closely related free-living organisms[15] but also in more distantly related Alveolata, such as dinoflagellates[16,17] (Supplementary Fig. 1). Thus the apical complex appears to be an ancient structure, of which the molecular composition, high-resolution

[1]Department of Cell Biology, University of Texas, Southwestern Medical Center, Dallas, TX, USA. [2]Department of Pharmacology, University of Texas, Southwestern Medical Center, Dallas, TX, USA. [3]Department of Biochemistry, University of Texas, Southwestern Medical Center, Dallas, TX, USA. [4]Present address: Department of Biophysics, University of Texas, Southwestern Medical Center, Dallas, TX, USA. ✉e-mail: michael.reese@utsouthwestern.edu; daniela.nicastro@utsouthwestern.edu

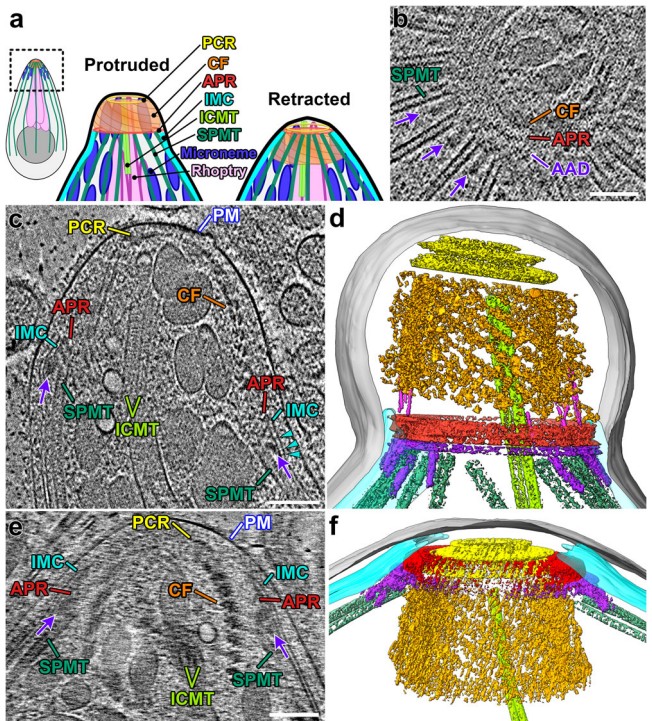

**Fig. 1 | Three-dimensional in situ architecture of the apicomplexan invasion machinery revealed by cryo-FIB milling and cryo-ET. a** Cartoon overview of the components of the coccidian apical complex comparing the protruded and retracted states. This coloring scheme will be used throughout the manuscript. **b** Tomographic slice through a partially retracted conoid that was cryo-FIB milled in cross-sectional orientation, clearly shows the SPMTs ending near the AAD ring and with AAD projections (purple arrows) interspersed between neighboring SPMTs. **c** Tomographic slice of the reconstructed apical end of *N. caninum* with protruded conoid. Note the density connecting the two membranes of the inner membrane complex (IMC, cyan arrowheads). For other labels and coloring, see below. **d** 3D segmentation and visualization of the apical complex from a protruded conoid (different tomogram from (**c**)). Labels and colors used throughout the manuscript unless otherwise noted: AAD (purple), amorphous APR-associated density ring and projections; actin-like filaments (magenta in (**c**)); APR (red), apical polar rings; CF (orange) conoid fiber, ICMT (light green) intraconoidal microtubules, IMC (cyan) inner membrane complex, PCR (yellow) pre-conoidal rings, PM (gray) plasma membrane, SPMT (dark green) subpellicular microtubules. **e** Tomographic slice of the reconstructed apical end of a milled *N. caninum* with a retracted conoid, annotated as in (**c**). **f** 3D segmentation and visualization of a retracted conoid (different tomogram from (**e**)) and colored as in (**d**). In longitudinal views, an apical tip is oriented towards the top of the images throughout the manuscript, unless otherwise noted. Scale bars: 100 nm (in **b**–**e**).

structure, and mechanistic understanding of its functions are still largely a mystery.

Common ultrastructural characteristics of the Alveolata superphylum include the cytoskeleton-supported vesicular structures that lie just basal to the plasma membrane and are known as alveoli[18]. In apicomplexans, these structures are called the inner membrane complex (IMC), which runs along the length of the parasite[19–22]. Apicomplexan parasites have two distinct sets of specialized organelles, called the micronemes and rhoptries, from which invasion factors and effector proteins are secreted (Fig. 1a)[23–25]. Whereas the rhoptry secretion requires close contact with the host cell plasma membrane, micronemes are thought to secrete continuously while the parasites are extracellular. Although these secretory organelles are not broadly conserved among alveolates, recent work has demonstrated that secretion from the rhoptries is mediated by a complex that is conserved both in its structure and protein components in ciliates such as *Tetrahymena* and *Paramecium*[26,27].

Cellular cryo-electron tomography (cryo-ET) is a powerful imaging technique that can reveal structural details with a molecular resolution, but the suitable thickness of biological samples is limited to a few hundred nanometers[28]. Most intact eukaryotic cells are thicker than this. Therefore, previous cryo-ET studies of whole eukaryotic cells have imaged either naturally thin cell regions, such as lamellipodia[29] and cilia[30], or cells that were compressed due to embedding in a thin layer of ice (compression force by water surface tension)[31,32]. Thicker specimens can be made thin enough for cryo-ET either by frozen-hydrated sectioning using a knife—which is prone to cutting artifacts[33]—or by cryo-focused ion beam milling[34].

We have applied cryo-ET to the coccidian apicomplexan parasites *Toxoplasma gondii*, arguably the most widespread and successful parasite in the world, and its close relative *Neospora caninum*. Our study mainly used cryo-focused ion beam (cryo-FIB) milling of unperturbed cells embedded in thick ice to compare the coccidian apical complex in its protruded and retracted states in situ, without the compression and deformation artifacts that have plagued studies of unmilled intact apicomplexan samples[31,32,35]. Our tomographic reconstructions provide an unrivaled view of the apical complex structure. Subtomogram averaging of the conoid fibers allowed us to clearly demonstrate that, contrary to a popular hypothesis[5,35], the conoid is not spring-like, and does not deform during cycles of protrusion and retraction. We also observed filaments that extend between the conoid and the apical polar rings (APR) through which the conoid moves, suggesting that polymerization of actin, or an actin-like protein, may play a role during conoid motion. Our data also provide an unprecedented view of the interactions between the parasite's secretory organelles and the apical complex cytoskeleton. Finally, we were able to capture the act of microneme fusion with the intact plasma membrane. Together, this work provides structural and mechanistic details for the apicomplexan invasion machinery, the apical complex. Because this complex is essential for apicomplexan parasite infection, understanding the molecular basis of its function could reveal important new targets for therapeutic intervention for some of the world's most devastating diseases.

## Results

### Cryo-FIB-milling followed by cryo-electron tomography preserves the 3D structure of the parasite cytoskeleton

The coccidian cytoskeleton appears remarkably well-preserved after detergent extraction and negative staining, which has led to many TEM studies of apicomplexan ultrastructure using this preparation method[1,4,5,36]. We first tested whether cryo-ET of detergent-extracted and then plunge-frozen *Toxoplasma* cells would allow us to gain an accurate 3D view, with improved resolution of the cytoskeletal structures, as compared to conventional TEM of negative-stained (dried) samples. Although cryo-ET of detergent-treated samples reveal cytoskeletal assemblies such as the apical conoid and subpellicular microtubules at high contrast (Supplementary Fig. 2), the tomograms also show artifacts from the detergent extraction as well as structural distortions (e.g., flattening; compare Supplementary Fig. 2e–h) that complicated gaining reliable structural information from these samples.

Therefore, to obtain the least perturbed and highest quality samples, we plunge-froze intact and live parasites in a relatively thick layer of ice (>1-μm thick; to avoid cell flattening by water surface tension). We then used cryo-FIB milling to generate 150–200 nm-thick lamellae of the vitrified, but otherwise native, parasites (Supplementary Fig. 3). To generate these samples, we used the NC1 strain of *Neospora caninum*, which is a BSL1 organism closely related to the human pathogen *Toxoplasma gondii* (Supplementary Fig. 1). The conoid of extracellular parasites protrudes and retracts continuously—with and without host cells present, but just before plunge-freezing, the parasites were either prepared in an intracellular-like buffer[37] or

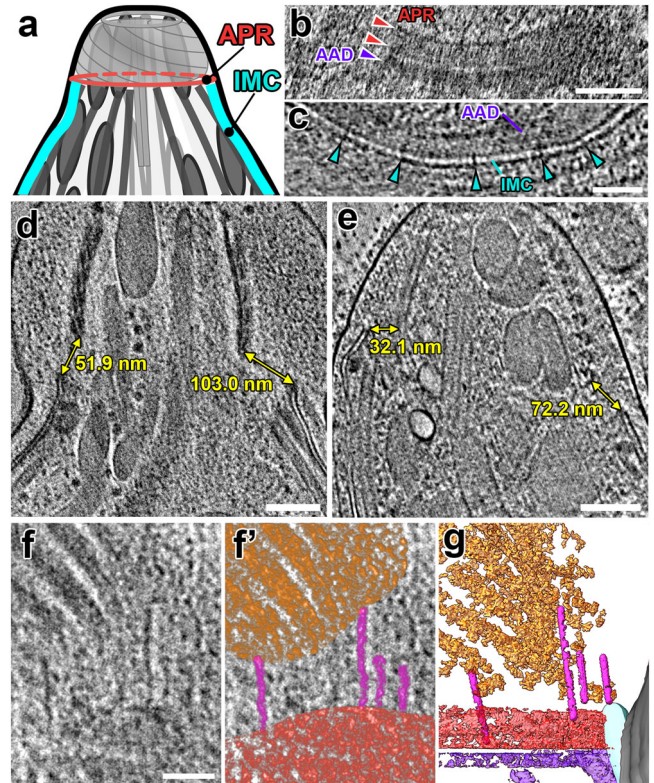

**Fig. 2 | The protruded conoid is tilted and off-center relative to the APRs.**
**a** Cartoon of the apical complex in the protruded state, highlighting the APR and IMC structures. **b** A tomographic slice (longitudinal orientation) shows two distinct APR rings (red arrowheads) and the AAD ring (purple arrowhead), which is located between the APR and IMC. **c** A cross-sectional tomographic slice shows the IMC near the apical edge with "spacer" densities (cyan arrowheads) between the two membranes. **d, e** Tomographic slices through the protruded conoid complex from two parasites. Measurements mark the minimum distances between the IMC and the conoid fibers on each side of the conoid. **f, g** Tomographic slice (**f**: original; **f**: pseudo-colored) and 3D-segmented isosurface rendering (**g**) of the same region at the base of a protruded conoid show filamentous actin-like densities (magenta) connecting between the conoid (orange) and the APRs (red). Scale bars: 100 nm (in **b**, **d**, **e**); 50 nm (in **c**, **f**).

incubated with 10 μM calcium ionophore for 10 min. so that the majority of parasites have conoids, preferably in the retracted or protruded states[6], respectively, without blocking conoid motility or cellular functions like secretion. The resulting cryo-tomograms reveal well-preserved structural details of the native *N. caninum* apical complex, including membranes, cytoskeletal assemblies, and organelles (Fig. 1 and Supplementary Movie 1).

### Structural organization of the apical cell region

We compared the architecture of the *N. caninum* apical complex in two states: conoid protruded and retracted. In the protruded state (Fig. 1c, d and Supplementary Movie 1), the pre-conoidal rings (PCRs) are the cytoskeletal structure closest to the apical plasma membrane (Fig. 1c, d), followed by the associated conoid that consists of 14–15 conoid fibers arranged in a spiral (Supplementary Fig. 2g, h). As compared to the protruded conoid state, the conoid fibers in the retracted state are tucked just basal to the APRs, and the PCRs are located slightly apical of the APR structures (Fig. 1e, f). Basal to the protruded conoid, the inner membrane complex (IMC) is clearly visible as a flat double membrane structure with associated densities just below the parasite plasma membrane (Fig. 1c, e). Notably, our in situ cryo-ET reveals several (non-periodic) densities that bridge between the two membranes of the extended sheet-like IMC (cyan arrowheads in Figs. 1c, 2c),

perhaps serving as "spacers" that constrain the distance between the membranes. The apical rim of the IMC sheet is attached to the apical polar rings (APRs; Fig. 1c–f). Just beneath the IMC, about 22 sub-pellicular microtubules extend basally from the APRs (Fig. 1b–e). Although the subpellicular microtubules and associated microtubule inner proteins (MIPs) are well-resolved in our cryo-tomograms, we have not focused on them here, because they have previously been analyzed in detail[35,38].

Super-resolution light microscopy has suggested components of the APR segregate into independent rings[39], but these rings have been difficult to resolve with negative staining EM. In our tomograms, we can clearly resolve several ring-shaped structures near the apical rim of the IMC: two distinct APRs with similar diameters, i.e., the A1 ring (apical) and the thicker A2 ring (basal) that may consist of two closely stacked sub-rings A2a and A2b (Fig. 2a, b and Supplementary Figs. 2d, 4a–c). We also observe a ring of amorphous density (named here "amorphous APR-associated density" or AAD) that is located between the IMC and the tips of the subpellicular microtubules (purple arrows/structure in Fig. 1b–f and Supplementary Figs. 2a, d, 4a–c). The AAD appears to be connected to the APR rings and has basal projections (21 nm wide and 64 nm long) that are sandwiched between adjacent subpellicular microtubules (Fig. 1b). The AAD projections have not been reported from conventional EM studies, but have recently been observed in another cryo-ET study (called "interspersed pillars" in ref. [35]). We observe the APRs, AAD ring, and AAD projections in tomograms of both the protruded and retracted conoid states. Interestingly, the AAD ring and projections appear preserved during detergent extraction of the parasite (Supplementary Figs. 2a, d, 4a, b). This localization and biochemical behavior suggest that the AAD ring and projections may contain components of the IMC apical cap, which have recently been localized by super-resolution light microscopy as interleaved between microtubules[4,40]. However, in these studies, known apical cap proteins such as ISP1 and AC9 extend ~1 μm from the APR, far beyond the ~64 nm occupied by the AAD projections. Therefore, the AAD ring and projections appear to be composed of proteins that have yet to be identified or precisely localized.

Surprisingly, and in contrast to the idea that the APR serves to strongly anchor the conoid to the parasite membrane, the protruded conoid never appears fully square to the APRs in our tomograms (compare protruded in Figs. 1c, d, 2d, e to retracted in Fig. 1e, f and Supplementary Fig. 4d–f). Instead, the conoid appears able to tilt and be off-center relative to the annular APR as it protrudes. To quantify this observation, we compared the distances between the apical rim of the IMC and the basal edge of the conoid from opposing sides in central slices through our tomograms (Fig. 2d, e and Supplementary Fig. 4d–f). We found that the difference between the opposing distances varied more in tomograms of the protruded state ($\Delta d = 52 \pm 12$ nm; $n = 3$) than in the retracted state ($\Delta d = 10 \pm 4$ nm; $n = 3$). These data were consistent with the idea that the conoid is "leashed" to the APRs and/or the apical rim of the IMC with the AAD by flexible and perhaps dynamic structures. Indeed, we frequently observed filaments between the APR region and the conoid fibers (Figs. 1d, 2f, g and Supplementary Fig. 4g–l).

Previous studies showed that apicomplexan "gliding motility" is driven by the treadmilling of actin fibers between the IMC cytoskeleton and the parasite plasma membrane[41,42]. Recently, F-actin nanobodies were used to demonstrate that actin fibers nucleate at the conoid and are passed down the parasite as it moves[3]. We clearly observe filaments consistent in diameter with actin fibers (~8 nm diameter) extending from the conoid to connect with the APR region, near where the hand-off to the IMC-associated myosin network[41,43] would be expected to occur. These filaments vary in length from 38–164 nm in our tomograms ($n = 14$ fibers from three tomograms), suggesting they are dynamic in nature (Figs. 1d, 2f–g and Supplementary Fig. 4g–l).

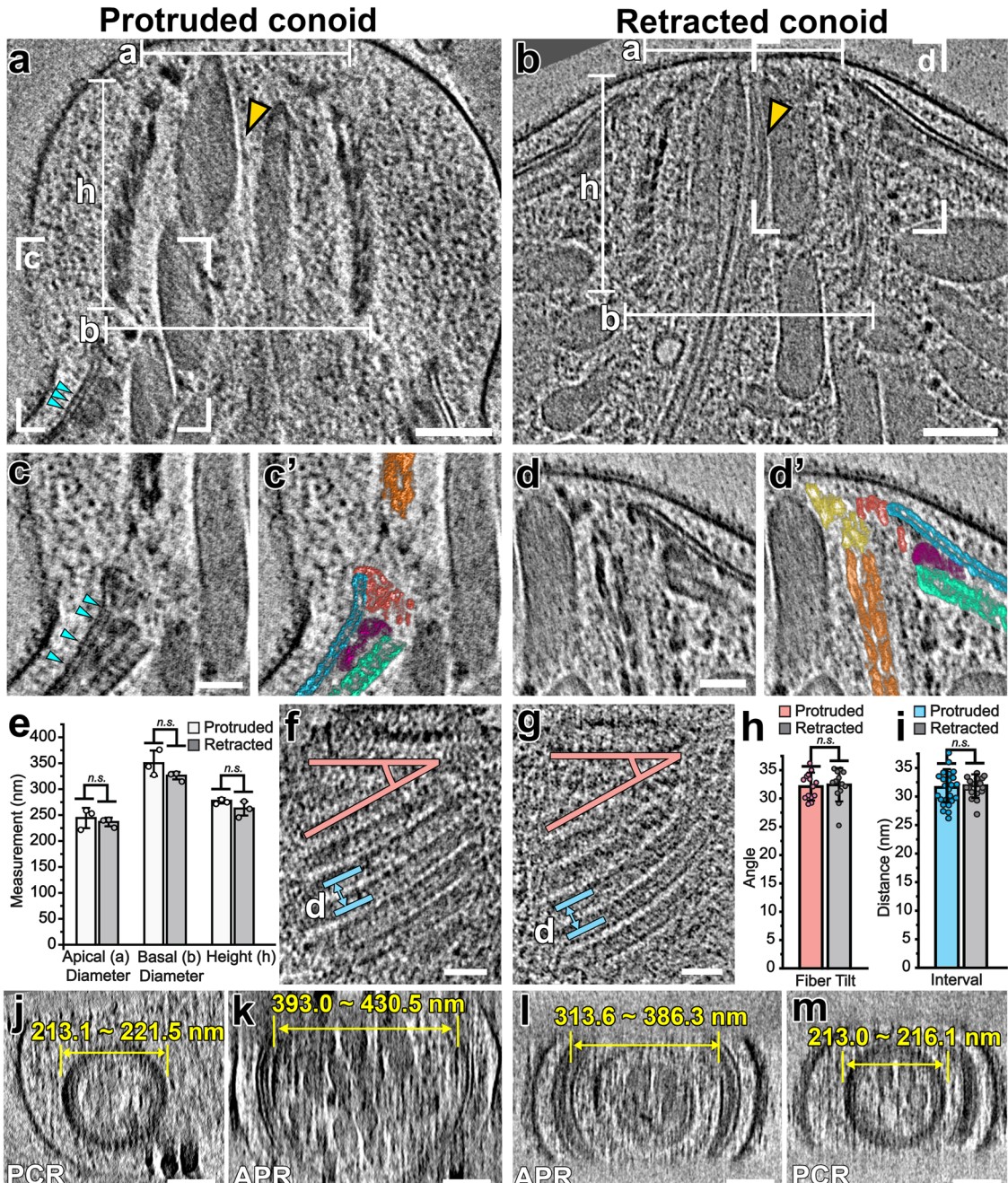

**Fig. 3 | The overall structure of the conoid remains unchanged during the cycles of protrusion and retraction. a–d'** Tomographic slices through the center of the apical complex show the conoid in the protruded (**a**) and retracted (**b**) states. White lines indicate the measurements of the apical diameter (*a*), the basal diameter (*b*), and the height (*h*) of the conoid structure. White boxes in (**a**, **b**) indicate the areas magnified in (**c**, **d**) where the conoid and the APR-associated complex interact; (**c'** and **d'**) are the pseudo-colored versions of (**c** and **d**), colored as detailed in Fig. 1. Note the elongated, sheet-like density (gold arrowheads in **a** and **b**) tracking between micronemes and ICMT inside the conoid. IMC "spacer" densities are indicated with cyan arrowheads in (**a**, **c**). **e** Measurements of the apical diameter, the basal diameter, and the height of protruded (*n* = 3 tomograms, white bars) and retracted (*n* = 3 tomograms, gray bars) conoids show no significant changes during the cycles of protrusion and retraction. **f–i** Tomographic slices (**f**, **g**) through the edge of reconstructed conoids (showing the conoid fibers in the longitudinal

section) in the protruded (**f**) and retracted (**g**) states. Lines indicate the measurement of the relative angle between the CFs and the PCR plane (*n* = 14 measurements for the protruded and 13 measurements for the retracted states, red lines), and the measurement of the distance between neighboring CFs (*n* = 24 measurements for the protruded and 18 measurements for the retracted states, blue lines); the results of latter measurements for protruded and retracted conoids are shown in (**h**) and (**i**), respectively. **j–m** Tomographic slices show representative PCRs (**j** and **m**) and APRs (**k** and **l**) in cross-sectional views of the apical complex in protruded (**j**, **k**) and retracted (**l**, **m**) states. Diameters of APRs and PCRs are indicated as ranges from all available tomograms (*n* = 3 for APRs in both states; *n* = 3 PCRs protruded; *n* = 2 PCRs retracted). Scale bars: 100 nm (in **a**, **b**, **j–m**); 50 nm (in **c**, **d**, **f**, **g**). Data were expressed as mean ± standard deviation. Statistical significance was calculated by a two-tailed Student's *t*-test. ns not significant (*p* > 0.05).

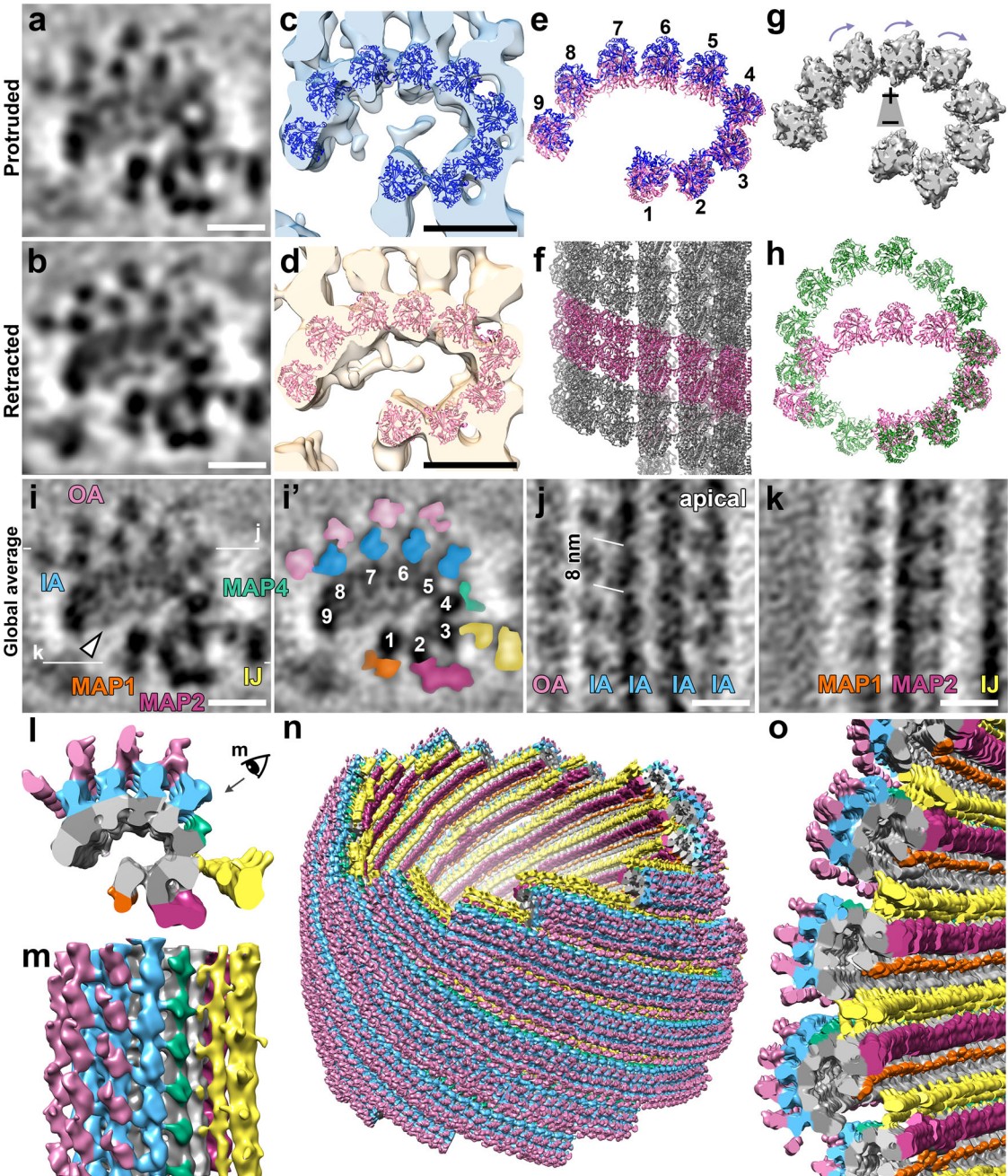

**Fig. 4 | Subtomogram averages of the conoid fibers show a C-shape architecture with nine tubulin protofilaments and associated proteins. a, b** Cross-sectional tomographic slices of the averaged 8-nm repeats of the CF fibers in the protruded (**a**) and retracted (**b**) states. **c**–**e** The high-resolution structure of tubulin was fitted into the subtomogram averages of the protruded (**c**, blue) and the retracted (**d**, pink) states. Comparison (**e**) of the two pseudo-atomic protofilament models in the protruded (blue) and retracted (pink) states shows no significant difference. **f** Longitudinal views of the pseudo-atomic protofilament model in the retracted state. The pitch was estimated based on the rise of the periodic CF-associated MAPs between neighboring protofilaments. **g** Isosurface rendering of the nice-fold averaged protofilaments display a "clockwise skew" when viewed from the conoid base, suggesting the minus ends of the conoid fibers are orientated to the base. **h** The arrangements of the modeled protofilaments in the CFs (pink) compared with the high-resolution cryo-EM structure of the typical 13-protofilament subpellicular microtubules (green; EMDB: EMD-23870). The most

substantial difference is the angle change between PF4 and PF5, causing a tight kink in the protofilament arrangement. **i**–**k** Tomographic slices of the global CF subtomogram averages that combine all data from both the protruded (**a**) and retracted (**b**) states viewed in cross-sectional (**i**: original; **i'**: pseudo-colored) and longitudinal (**j** and **k**) orientations. The white lines in (**i**) indicate the locations of the slices in the respective panels. Labels and coloring see below. **l**–**o** Isosurface renderings show the 3D structures of the averaged CF repeats in cross-sectional (**l**) and longitudinal (**m**) views, as well as the CFs from a complete conoid (**n**) by assembling the averaged 8-nm repeats back into the full tomogram. This and the zoom-in (**o**) show that the open face of the C-shaped CFs faces the interior of the conoid. Labels and coloring: 1–9, protofilaments; IA (blue) and OA (pink) "inner-layer arm" and "outer-layer arm" densities, IJ (yellow) inner junction, MAP; MAP1 (orange), MAP2 (magenta), MAP4 (green), microtubule-associated proteins. Scale bars: 10 nm (in **a**, **b**, **i**–**k**).

In contrast to intact parasites, in our detergent-extracted samples, the conoid base appears to sit directly on the APRs (Supplementary Fig. 2a, 4a, b), as is typically seen in similarly treated samples by negative staining EM[4,5,36]. This collapse of the gap and filamentous structures incorrectly suggests that the APR and its connections to the conoid are much more rigid than we observe in our cryo-FIB-milled, but otherwise unperturbed, samples, which again highlights the value of our in situ analysis of the structures in their native states.

## Conoid structure is unchanged during cycles of protrusion and retraction

As described below, the conoid fibers are unusual tubulin polymers in which the protofilaments form an open C-shaped cross-section (Supplementary Movie 1). These fibers were initially described as a wound in a "spring-like" structure[5], leading to a model that the conoid motions were driven by a spring-like mechanism, involving cycles of deformation in the conoid structure followed by release. This hypothesis is made more attractive because of the conoid fibers unusual C-shape, which would be expected to be more deformable/compressible than closed microtubules. Prior to the development of cryo-FIB milling, electron microscopy and tomography analysis of the conoid required detergent extraction of the parasite membrane and/or cell flattening, which often result in structural artifacts during sample preparation and during image processing, where orientation bias prevents proper missing-wedge correction (Supplementary Fig. 2a-f, i). Our data of the unperturbed, native apical complex allows us to unambiguously address changes of the conoid ultrastructure during different functional states.

We, therefore, sought to assess whether the conoid movements behaved according to the spring-like model involving the deformation of the conoid and conoid fibers. We reasoned that any gross ultrastructural change in the conformation of the conoid polymer would result in changes in the overall dimensions and architectures of the conoid. We compared the dimensions of protruded and retracted conoids in tomograms from cryo-FIB-milled *N. caninum* (Fig. 3 and Supplementary Fig. 5). Neither the apical conoid diameters ($244 \pm 19$ nm protruded vs. $236 \pm 8$ nm retracted), the basal diameters ($350 \pm 25$ nm vs. $326 \pm 7$ nm), nor the conoid heights ($277 \pm 5$ nm vs. $262 \pm 13$ nm) were significantly different between the two states (Fig. 3e and Supplementary Fig. 5a). Similarly, the angle of the conoid fibers relative to the PCRs, and the intervals between adjacent conoid fibers were indistinguishable in the two states (Fig. 3f–i). Furthermore, both the mean conoid fiber length and the distribution of lengths in the population were not significantly different between the protruded and retracted states ($399 \pm 37$ nm vs. $405 \pm 45$ nm; Supplementary Fig. 6a).

Whereas the apical tip of the conoid and the PCRs are levels with the APR when the conoid is retracted, after an extension of the conoid, the base of the conoid is located in the region of the APR (compare Fig. 3a, c, c' with 3b, d, d'). In switching between the protruded and retracted states, the apical IMC regions, the AAD, and the APRs all change their position relative to the conoid and its associated PCRs, and to the plasma membrane (Fig. 3a–d'). The positions of the APRs and AAD appear closely coupled to the apical rim of the IMC in our tomograms, which appear to flex apically in the protruded state. We, therefore, asked whether the APRs exhibited structural changes correlated with conoid protrusion and retraction. We measured the diameters of the APRs in each of our tomograms and determined that there was a striking 10–30% increase in APR diameter in the protruded versus retracted states (Fig. 3k, l and Supplementary Fig. 5b). In contrast to the APRs, the structure of the PCRs and conoid showed no observable differences between the two states (Fig. 3j, m and Supplementary Fig. 5c).

## Molecular structure and polarity of the conoid fibers

We next assessed the conoid fiber structure at a higher resolution by generating separate subtomogram averages of the conoid fibers from the protruded (Fig. 4a; ~3.0 nm resolution at 0.5 FSC, see Supplementary Fig. 6b) and retracted states (Fig. 4b; ~3.2 nm resolution at 0.5 FSC, see Supplementary Fig. 6b). Visually, the subtomogram averages of the conoid fibers from either state appear largely indistinguishable, and reveal the nine tubulin protofilaments[5], as well as distinct protein densities decorating the fibers (Fig. 4a, b). To better assess any variation between the two states, we fitted tubulin dimers into the two averaged cryo-ET maps using Chimera[44] (Fig. 4c, d) with correlation coefficients of ~0.9. We then overlaid the best-fit models of a single repeat of the conoid fiber from each state (Fig. 4e). Consistent with the lack of difference between the shape of the entire conoid structure between the two states (Fig. 3), each of the nine protofilament subunits appears positioned indistinguishably between the two states within the resolution of the data and error of the fit (Fig. 4e). Note that the three protofilament subunits with the highest RMSD in their overlay (PF5-7; Supplementary Fig. 6c) are fitted in a lower resolution region of the averages.

Taken together, our data demonstrate that the conoid appears to move as a rigid body during protrusion and retraction, rather than as a conformationally deformable structure, and there is no obvious rearrangement of protofilaments within the conoid fibers during this process. From these data, we cannot, however, rule out minor conformational changes between the two states. Our in situ data also corroborate that the conoid fibers form an unusual open C-shaped arrangement of nine protofilaments[5] that requires contacts distinct from a typical 13-protofilament MT (Fig. 4h).

Having demonstrated that the conoid fibers do not undergo major conformational changes between the protruded and retracted states, we combined particles from both states to calculate a global average of the conoid fibers (Fig. 4i–k). The resulting average of the two states shows a markedly improved signal-to-noise ratio and a resolution of ~2.8 nm at the 0.5 FSC criterion (or ~2.2 nm at the 0.143 FSC criterion; Supplementary Fig. 6b) and reveals density coating both the external (MT-associated proteins; MAPs) and internal (MT inner proteins; MIPs) edges of the conoid fiber (Fig. 4i–o and Supplementary Movie 1). The MIP densities bridge multiple protofilaments and coat almost the entire inner surface of the conoid fiber (Fig. 4i, i'). These contacts may help explain the apparent rigidity of the conoid fiber structure. The external surface of the conoid fibers also appears almost entirely decorated with MAPs (Fig. 4i–o and Supplementary Fig. 2i–k). Associated with PF5-8 are two layers of MAPs, named here as "inner-layer arm" and "outer-layer arm" densities (IA and OA; Fig. 4i, j and Supplementary Fig. 2i, j). Additional MAPs include MAP1, 2, and 4 which are associated with PF1, 2, and 4, respectively. We also observe inner-junction MAP densities (IJ) that connect PF3 of one conoid fiber (*n*) with PF9 of the neighboring fiber (*n* + 1; Fig. 4i, i', k–o and Supplementary Figs. 2i, i', k, 6f–h). These inter-fiber contacts likely stabilize the spiral conoid structure and reinforce its rigidity. Most MAPs exhibit a clear 8 nm periodicity along the length of the conoid fiber (Fig. 4j, k, m and Supplementary Figs. 2j, k, 6e, e').

Regular microtubules are helical assemblies that have a characteristic handedness, helical pitch, and polarity. For example, a typical 13-protofilament microtubule forms a straight polymer with a rise of three monomers (12 nm) per left-handed helical turn (a "13-3 helix"), or 0.92 nm rise per protofilament-to-protofilament contact[45]. Using the regularly spaced MAPs as a guide, we were able to assign the pitch of the conoid fiber in longitudinal views (Fig. 4f, j, k and Supplementary Figs. 2j, 6e, e'). As the C-shaped conoid fibers are formed from an open, rather than closed-helical microtubule, we calculated only the protofilament-to-protofilament pitch. Our model indicates that the conoid fiber forms through a left-handed assembly−like a typical microtubule, but has an estimated pitch of ~1.5 nm protofilament-to-protofilament, which is ~1.6-fold larger than a typical microtubule

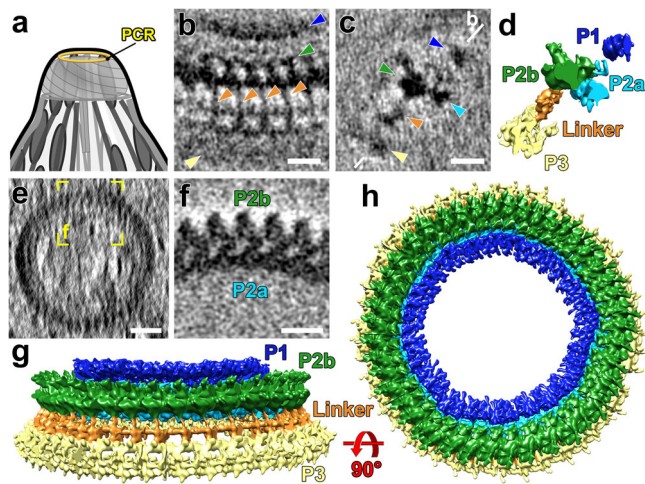

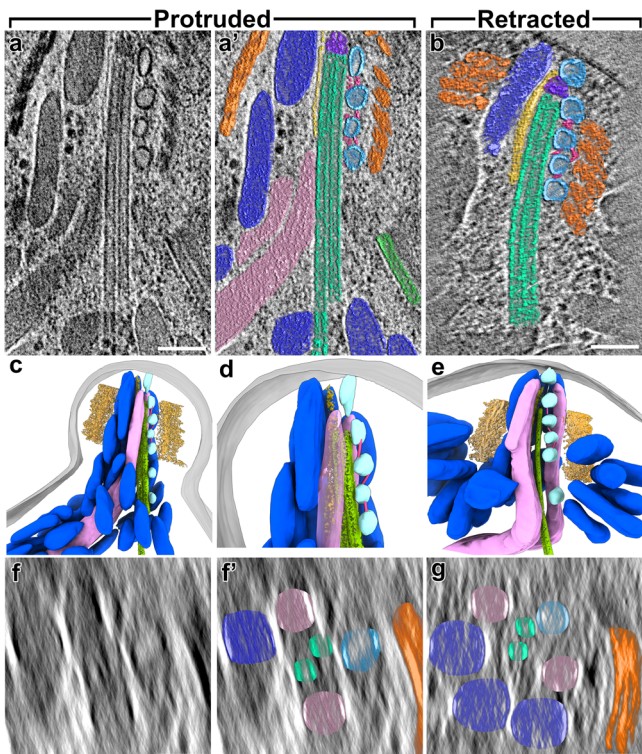

**Fig. 5 | Subtomogram averages of the PCRs show three rings and a linker.**
**a** Cartoon of the apical complex highlighting the location of the PCRs.
**b**–**d** Tomographic slices (**b** and **c**), and isosurface rendering of the averaged PCR repeats (**d**) viewed in the tangential (**b**) and longitudinal (**c** and **d**) orientations, showing different components of the PCR including the apical P1 (highlighted by blue arrowheads), P2a (cyan), P2b (green), P3 (yellow), and the linker (orange) between P2b and P3. The white line in (**c**) indicates the location of the slice in (**b**).
**e**, **f** Cross-sectional slices from a raw tomogram (**e**) and the averaged PCR repeats (**f**) show that the round PCR P2 ring is composed of an outer and an inner ring, P2a and P2b, respectively. The yellow square in (**e**) indicates the orientation and location of the subtomogram average displayed in (**f**). **g**, **h** Isosurface renderings show the complete PCRs by assembling the averaged repeats back to the full tomogram. Scale bars, 20 nm (in **b**, **c**, **f**); 50 nm (in **e**).

**Fig. 6 | The apical secretory machinery is organized around the intraconoidal microtubules (ICMT) in both protruded and retracted states. a**, **b** Tomographic slices show the apical tip of a *N. caninum* cell with the secretory organelles organized around the ICMT (green) inside the protruded conoid complex in longitudinal views (**a**, **a'**–protruded original and pseudo-colored; **b**–retracted). The long ICMTs connect the apex of the conoid to the cytosol, and are closely co-localized with the secretory vesicles (light blue) and two rhoptries (rose). Note that the membrane-associated, most-apical vesicle is not visible in the tomographic slice shown in (**a**), but is visible in panels (**c**, **d**) and Fig. 7b. Other coloring: CFs (orange), micronemes (dark blue), sheet-like density (gold), inter-vesicular connections (pink), "crowning" density that caps the apical minus-end of the ICMT (purple). **c**–**e** 3D segmentation and visualization of tomograms shown in (**a** and **b**, respectively), i.e., with protruded (**c**, **d**) and retracted (**e**) conoid, showing the overall organization of secretory organelles within the conoid complex, including CFs (trimmed from the front to show the content inside), micronemes, rhoptries, ICMT, vesicles, inter-vesicular connections, sheet-like structure along micronemes, and plasma membrane (gray). **d** shows a zoom-in from (**c**), with conoid fibers hidden for clarity. **f**, **g** Cross-sectional slices through the protruded (**f**, **f'**–original and pseudo-colored) and retracted (**g**) conoids from the tomograms shown in (**a** and **b**). Scale bars: 100 nm (in **a**, **b**); 50 nm (in **f**, **g**).

(Fig. 4f and Supplementary Fig. 6e, e'). To determine the structural polarity of the conoid fiber, we performed a ninefold average of the protofilaments (using the tangential/normal to the C-shape curvature of the fiber). The isosurface rendering of the ninefold averaged protofilaments displayed a "clockwise skew" when viewed from the conoid base, suggesting the minus ends of the conoid fibers are located at the base of the conoid complex (Fig. 4g).

### Subtomogram averages of the pre-conoidal rings reveal three layers of periodic densities that are connected
Instead of the previously reported two PCRs[5,9], we resolve three PCRs (Fig. 5 and Supplementary Figs. 2b, 7): P1 is the most apical and smallest ring (Fig. 5b–d, g, h and Supplementary Fig. 7e, i) with 151 ± 6 nm diameter—which was likely overlooked in previous detergent-treated and negatively stained samples; P2 (middle; Fig. 5b–h and Supplementary Fig. 7f, j) consists of two sub-rings with the following diameters: P2a—178 ± 2 nm, P2b—217 ± 8 nm; and P3 (basal; Fig. 5c, d, g, h and Supplementary Fig. 7g, k) has a diameter of 258 ± 8 nm (Fig. 5b–h). All diameters were measured on the outside edges of the rings in 4 different cells (*n* = 4). Subtomogram averaging of the PCR from four reconstructed *N. caninum* cells produced a 4.9 nm resolution average (0.5 FSC criterion, Supplementary Fig. 6b), and revealed both the periodicity and connections between the PCR layers. In our tomograms, each of the layers appears to have the same periodicity, with 45–47 subunits per ring (Fig. 5g, h). P2 and P3 are connected with regularly spaced linkers that are ~25 nm long, with apparent bridging density between the neighboring linker subunits (Fig. 5b–d, g and Supplementary Figs. 2b, 7c).

### In situ cryo-ET of native cells reveals functional segregation of secretory organelles within the coccidian conoid
Conventional EM of chemically-fixed and resin-embedded apicomplexan parasites has provided a basic overview of the organization of

the apical secretory organelles within the coccidian conoid[5,46], which we used to guide our analysis of the in situ structure of the apical complex—including the secretory organelles —in our tomograms (Figs. 6, 7, Supplementary Fig. 8, and Supplementary Movie 1). We clearly observe the central pair of intraconoidal microtubules (ICMT), extending from the conoid tip basally for 500–800 nm (Fig. 6a, b and Supplementary Fig. 6i), which is longer than the 350 nm previously reported[5]. The ICMT have been proposed to be a major organizing center for secretion in coccidia[47,48], and appears to have a set of associated proteins that are distinct from both the subpellicular microtubules and conoid fibers[5], though only one such protein has been identified to date[49].

In line with what has been observed with EM of resin-embedded samples[5], we observe that distinct secretory organelles (the rhoptries and micronemes) segregate to opposite sides of the ICMT in tomograms from both protruded and retracted conoids (Fig. 6). We also

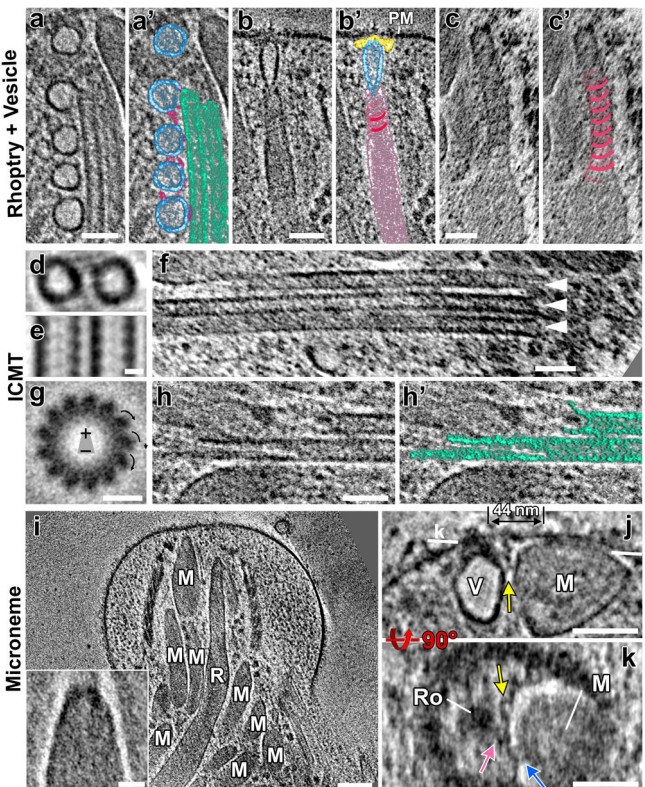

**Fig. 7 | The intraconoidal microtubules and functional segregation of secretory organelles within the conoid complex revealed by cryo-FIB milling and cryo-ET. a–c'** Tomographic slices (**a–c**: original; **a'–c'**: pseudo-colored) show: (**a, a'**) five regularly spaced vesicles (light blue), of which four are tracking along one of the microtubules of the ICMT (light green) and are connected by inter-vesicular linkers (pink); (**b, b'**) a rhoptry (rose) interacting with the plasma membrane (PM) via the most-apical vesicle (light blue) and the "rosette" docking complex (yellow); and (**c, c'**) a spiraling scaffold (dark rose) associated with the rhoptry membrane in the rhoptry-neck region. **d, e** Slices of the subtomogram averaged 8-nm repeats of the ICMTs viewed in cross-sectional (**d**) and longitudinal (**e**) orientations. **f** Occasionally, more than two microtubules were observed in the ICMT complex. Shown here is an example of three microtubules (white arrowheads). **g** Thirteenfold rotationally averaged ICMT from 53 subtomograms of detergent-extracted *Toxoplasma* cells. The arrows indicate the clockwise skew of the protofilaments when the ICMT are viewed from apical to basal, indicating that the minus ends of the ICMTs are oriented apically in the parasite. **h, h'** The basal ends of the ICMTs showed flared ends and different lengths of protofilaments, which is usually associated with dynamic plus-ends of MTs. **i** A tomographic slice of a protruded conoid shows the organization of micronemes. Insert: the subtomogram average of 25 microneme apical tips shows a flattened, electron-dense cap. **j, k** Tomographic slices provide a side (**j**) and a top cross-sectional view (**k**) of two secretory organelles, a microneme (M) and a rhoptry-associated vesicle (V), that are docked side-by-side to the plasma membrane (the vesicle through the rosette (Ro)), but with distinct docking sites (44 nm apart). Note that both organelles are tethered (pink and blue arrows) to the same plasma membrane-anchored ridge (yellow arrows). The white line in (**j**) indicates the location of the slice in (**k**). Scale bars: 100 nm (in **l**); 50 nm (in **a–c, f, h, j, k**); 20 nm (in **i** insert), 10 nm (in **d, e, g**).

observed four to six regularly spaced vesicles tracking along one of the microtubules of the ICMT (Figs. 6a–e, 7a, a'), consistent with previous reports[26,46]. We observed a single vesicle just apical of the ICMT that appears docked to the density of the recently described "rosette" (Fig. 7b-b'), and which has been associated with facilitating rhoptry secretion[26]. Consistent with this function, we observe that the most-apical, membrane-docked vesicle connects to the apical tips of one to two rhoptries (Fig. 7b, b' and Supplementary Fig. 8a). We also identified clear densities connecting the vesicles to both the ICMT and to each other within a chain (Figs. 6a–e, 7a). Notably, we did not observe

this bridging density at the apical, membrane-docked vesicle (Fig. 7a; $n = 6$ tomograms with the vesicle chain and apical vesicle present; note that in other tomograms, the apical vesicle was milled away by cryo-FIB). Whereas the apical vesicle likely originated from the ICMT-associated vesicle chain, in docking with the rhoptry and plasma membrane, it appears to have been released from the vesicle chain and its connections. We also observed that some of the vesicles appear elongated along the axis of the ICMT, suggesting that they are under tension and have been captured in active trafficking along the ICMT (Fig. 6a).

A parasite cell has ~10 rhoptries and tens of micronemes, though only a subset of either organelle is docked at the parasite conoid at any given time[46]. In all tomograms that contained the full diameter of the parasite conoid (6 of 9 tomograms), we observed two rhoptries tracking along the ICMT, but on opposite sides of the ICMT (Fig. 6c–g; colored rose). However, we do not observe clear linkers connecting the rhoptries to the ICMT. Within the apical region of the parasite, a spiral-shaped density corkscrews along the rhoptry membranes (Fig. 7b–c'), reminiscent of the localization pattern of Ferlin-2 seen by immuno-EM[50]. The filaments spiraling up the rhoptries within the conoid were previously estimated to have a diameter of ~33 nm and a pitch of ~21.5 nm from tomograms of unmilled samples in which the parasites had been flattened due to surface tension[51]. We found that such a predicted helix is consistent with our measurements of ~36 nm in diameter and ~17 nm in pitch from cryo-FIB-milled parasites (Fig. 7c, c').

Because of their short length and potential variation in their associated proteins along their length, subtomogram averaging of the ICMT did not yield a sufficient resolution to clearly resolve MAPs or MIPs (Fig. 7d, e). Instead, we examined the chirality of the ICMT cross-section[52] using an ICMT subtomogram average from a detergent-extracted tomogram (Fig. 7g). This analysis demonstrates that both of the paired ICMT, unlike the conoid fibers, have their minus ends oriented apically. Moreover, the basal, plus-end of the ICMT pair exhibits the splayed morphology typical of a dynamic microtubule (Fig. 7h). This observation suggests that the ICMT, unlike most other apicomplexan MT structures, exhibits dynamic instability at its plus-end (Fig. 7h), and may explain why the previously reported ICMT lengths vary largely between samples and are consistently shorter in extracted samples[5] versus those that we measure from intact, native parasites. We also observe a density that caps the apical, minus-end of the ICMT (Fig. 6a, b), which would be consistent with a microtubule-organizing center from which the ICMT are polymerizing. In coccidia, however, γ-tubulin appears to be restricted to the parasite centrioles and cytoplasm, and has not been localized to the apical complex[53]. As has been previously reported[46], some ICMT contained a third microtubule (Fig. 7f), suggesting that rigorous control of the structure is unnecessary for its function.

Notably, the ICMT and its associated vesicles and rhoptries segregate to one side of the conoid that is distinct from the region occupied by the micronemes (Fig. 6c–g); thus, the ICMT are not directly involved in organizing micronemes for secretion. Nevertheless, the micronemes do not appear disorganized (Figs. 6c–e; 7i). Instead, they are arrayed in clusters and show a distinct polarized orientation. The micronemes are relatively uniform in length ($220 \pm 30$ nm) and width ($58 \pm 11$ nm; Supplementary Fig. 8b–h), and the basal ends are rounded, whereas the apical ends appear narrow with a flattened, electron-dense cap (Fig. 7i; inset). This density suggests an undescribed scaffolding complex that we propose organizes the micronemes and assists in their trafficking. Furthermore, in one tomogram of a retracted conoid, we observed two micronemes that appear to be docked through their apical tips with the plasma membrane (Supplementary Fig. 8i). In addition, we consistently identified a long sheet-like density between the micronemes and the ICMT in our tomograms (Fig. 6a, b and Supplementary Movie 1). The sheet varies in thickness between about 4–8 nm, appears fibrous, is about 15–30 nm

wide, and can often be traced to extend from the top of the conoid to below its base. These data suggest that the sheet-like structure may assist in the organization and trafficking of secretory organelles.

Whereas micronemes secrete continuously to promote the motility of extracellular parasites, they are also intimately involved in triggering host cell invasion. To initiate an invasion of a host cell, all apicomplexan parasites rely on the secretion of a receptor/co-receptor pair comprising a microneme protein (AMA1) and a rhoptry proteins (RON2)[54,55]. Microneme and rhoptry components of the invasion apparatus have been proposed to mix as they were secreted through a common path at the conoid[23]. That we observe micronemes segregated from the docked rhoptries calls this model into question. Furthermore, in one tomogram of a protruded conoid, we identified a microneme that appears to have been captured in the process of plasma membrane docking/secretion (Fig. 7j, k and Supplementary Fig. 8j–m). Consistent with an ongoing secretion event, the microneme in question is about half the length of all other micronemes measured (Supplementary Fig. 8h, red data points). We observe a density consistent with one large or multiple contact sites in a circular region of ~85 nm in diameter (Fig. 7k and Supplementary Fig. 8j, l). We also observe an apparent bolus of density on the surface of the plasma membrane associated with the contact site (Fig. 7j and Supplementary Fig. 8j, m), consistent with the secretion of microneme contents. Notably, the docking site of this secreting microneme is 44 nm from the rosette and its docked rhoptry/apical vesicle, indicating that the two organelles secrete at separate sites using distinct secretory machineries (Fig. 7j). However, both the rhoptry-associated vesicle and the docked microneme are tethered to the same ridge (Fig. 7j, k and Supplementary Fig. 8j; yellow arrows) and membrane-anchor (Supplementary Fig. 8m and inset; yellow arrowheads), which would likely facilitate spatial and temporal coordination of secretion.

## Discussion

The conoid structure of the apical complex has captured the imagination of microscopists since its earliest descriptions by conventional TEM[56,57]. Here, we have applied cryo-ET to cryo-FIB-milled parasites to compare the native in situ structures of the coccidian apical complex cytoskeleton and associated secretory organelles in the protruded and retracted states of the conoid. A conoid protrusion is associated with striking morphological changes at the parasite apical tip that are visible by light microscopy[6,58]. These morphological changes, coupled with the conoid's unusual spiral shape, have led to a model that the conoid is spring-like and deforms during its movements (protrusion/retraction). Previous cryo-EM of detergent-extracted parasites[59] and a recently published cryo-ET analysis[35], document differences in protruded versus retracted conoids. However, subtomogram averaging in the Sun et al. study focused on detergent-extracted samples, and the samples used in both previous analyses were compressed, which can lead to structural artifacts.

In contrast, our cryo-FIB-milled samples preserve the circularity of the coccidian apical complex structure (Supplementary Fig. 2g, h and Supplementary Movie 1), which allows a more reliable interrogation of its native structure. Our data show that both the ultrastructure and molecular organization of the protruded and retracted states of the conoid are indistinguishable, indicating the conoid fibers do not deform like a spring during conoid movements, and therefore do not provide energy to assist in either conoid motion or in secretion.

Whereas we did not observe changes in the structure of the retracted versus protruded conoid, we found that the APR appears to dilate during protrusion which appears coupled with flexion of the IMC at the apical tip during protrusion. We also did not observe close contacts between the conoid and the APR, rather the protruded conoid appears often somewhat tilted relative to the APR plane, consistent with flexible and/or dynamic tethers connecting the conoid to the APR and/or the apical rim of the IMC. RNG2 appears to be one such protein,

as its N- and C-termini appear to span between the conoid and APR during protrusion[60]. Whereas we would not expect to identify the density of such an intrinsically disordered protein, we did observe densities that span between the base of the protruded conoid and the APR that were consistent with actin/actin-like filaments. Therefore, it is possible that actin polymerization at this site is at least partially responsible for generating the force of conoid motion. Notably, an early description of conoid protrusion found that depolymerization of actin with cytochalasin D attenuated, but did not completely abrogate, conoid protrusion[6]. Recently, Formin-1, which is responsible for the polymerization of actin at the conoid, has been demonstrated essential for conoid protrusion[61]. Nevertheless, further high-resolution structural studies using genetic perturbation will be required to place individual proteins in the apical complex and tease apart the physical basis of the conoid dynamics and function.

Because our cryo-FIB-milled samples preserved the parasite membranes, we obtained an unparalleled view of the interactions between the cytoskeleton of the apical complex and the parasite's specialized secretory organelles. We observed close contacts between the ICMT and both the parasite rhoptries and the apical vesicle chain. This suggests that the ICMT are responsible for organizing these organelles, but not the micronemes, with which we observed no direct contacts. Intriguingly, whereas the ICMT are not broadly conserved in Apicomplexa outside of Eucoccidiorida, a pair of ICMT have been reported in *Chromera velia*[15], a free-living alveolate that is among the extant organisms most closely related to Apicomplexa (Supplementary Fig. 1). Thus, the ICMT were likely present in the ancestral species. That the apical vesicles and rhoptries track along the ICMT suggests that a microtubule motor may be facilitating their organization, though parasite kinesins and dyneins have not been localized to the ICMT. Because the majority of trafficking in Apicomplexa appears to occur on actin filaments[3,62–64], the parasite microtubule motors have not yet been systematically characterized. Notably, only a single ICMT-localized protein has been identified to date[49], which lacks clear homology to known structures.

A major open question in the field is the molecular basis of the force that drives the secretion of the invasion-associated organelles docked at the apical complex. The rhoptries are >1 μm in length, and likely require a more intricate machinery than simple membrane docking to drive secretion. Intriguingly, depolymerization of actin using cytochalasin D blocks parasite motility and invasion but not host cell attachment and rhoptry secretion[65], suggesting actin is not responsible for driving secretion. Recent work identified the apicomplexan Nd proteins that comprise the "apical rosettes", structures best characterized in the ciliate group of Alveolata[27]. Similar to their role in ciliates, the apicomplexan Nd proteins are essential for rhoptry secretion[26], and the rosette structure was recently described using cryo-ET of unmilled parasites[51]. The *Toxoplasma* protein Ferlin-2 is also required for rhoptry secretion[50], and we and others[51] identified a density spiraling around the rhoptries within the conoid that is reminiscent of the published Ferlin-2 immuno-EM staining pattern[50]. These spiraling filaments may act somewhat like dynamin to squeeze the rhoptries during secretion. Among the *Toxoplasma* Nd-associated proteins were putative GTPase-related proteins[26], suggesting a dynamin-like activity may be present at this site as well.

With the ability to rapidly freeze and thus capture snapshots of highly dynamic cellular processes, we were able to observe a docked microneme that appears to be in the process of secretion. We found that the site of microneme secretion is distinct from that of a docked rhoptry-associated vesicle, suggesting that the components of the invasion machinery secreted by these two organelles must find each other in the apical plasma membrane after secretion. Our data indicate that microneme secretion can occur during conoid protrusion, events that have been indirectly correlated by other studies[7]. Note, however, that these data do not rule out additional secretion during

the retracted state. Finally, we not only observed a plasma membrane-associated ridge and anchor between the docked microneme and rhoptry-associated vesicle, but also an elongated (fibrous) sheet that tracks between the micronemes and the ICMT. These structures may represent cytoskeletal elements responsible for the organization, trafficking, and coordinated membrane docking of the apical secretory organelles. Microneme turnover requires trafficking along actin, though biogenesis and organization of micronemes appear actin-independent[63]. Further studies will be required to identify the molecular components of the anchor, ridge, and sheet, and their functions in organellar trafficking and secretion.

In summary, cryo-ET of cryo-FIB-milled parasites has enabled us to examine the apical complex in situ, overcoming the compression artifacts that occur when preparing intact cells in a thin layer of ice. These significant advances allowed us to interrogate the native structure of the conoid complex and its movements during retraction and protrusion. We were able to unambiguously demonstrate that the conoid moves as a rigid body during protrusion, with filamentous, actin-like projections that connect it to the APR. The next frontier in understanding the mechanics of the apical complex will be in capturing parasites during host cell invasion. It is possible that components of the apical complex will undergo some conformational change when the parasite is in intimate contact with a host cell membrane. We also anticipate that further studies combining proteomic data with cryo-ET of cryo-FIB-milled parasites will enable the placement of individual proteins into the apical complex structure, which will shed new light onto the molecular and structural basis of its movements and function.

## Methods

### Phylogenetic analysis
The phylogenetic tree in Supplementary Fig. 1 was estimated in RAxMLv8.2[66] using the LG substitution model with gamma rate heterogeneity and empirical frequencies with 1000 bootstrap using an alignment of the protein sequences for HSP90 concatenated with RPS11 (accessions: AFC36923.1, BESB_021480, XP_029219085.1, LOC34617734, XP_022591029.1, ETH2_0701200, ETH2_0910900, NCLIV_040880, XP_003884203.1, TGME49_288380, XP_002366350.1, SN3_03000005, SN3_01300510, PBANKA_0805700, XP_034421046.1, PF3D7_0708400, XP_001351246.1, PVP01_0108700, PVP01_0822500, GNI_014030, XP_011128490.1, KVP17_001483, KAH0483594.1, FG379_001268, KAH7649672.1, Chro.30427, OLQ16118.1, cgd3_3770, CPATCC_001922, Vbra_12473, CEM00719.1, Cvel_2184, Cvel_482, AAA30132.1, XP_764864.1, XP_952473.1, XP_952423.1, XP_001611554.1, XP_001609980.1, XP_002775585.1, XP_002766754.1, XP_001447795.1, XP_001445466.1, XP_001009780.1, XP_001030186.1, AAR27544.1, XP_009040431.1, XP_009033899.1).

### Cell culture and cryo-preparation
Human foreskin fibroblasts (HFF; a gift from John Boothroyd) were grown in Dulbecco's modified Eagle's medium supplemented with 10% fetal bovine serum and 2 mM glutamine. *Toxoplasma gondii* (RH strain) and *Neospora caninum* (NC1 strain) tachyzoites were maintained in confluent monolayers of HFF. Detergent-extracted *Toxoplasma* cells and the associated subtomogram averages are displayed in Fig. 7g and Supplementary Figs. 2a–f, i–k, 4a, b, 6i. Tomographic reconstructions of cryo-FIB-milled *N. caninum* cells and the corresponding subtomogram averages and data analyses are presented in Figs. 1–6, 7a–f, h–k and Supplementary Figs. 2g, h, 3, 4c–l, 5, 6a–h, 7, 8. For the preparation of parasites for cryo-ET, highly infected HFF monolayers were mechanically disrupted by passage through a 27 gauge needle to release the parasites. For "retracted conoid" samples, parasites were kept in "Endo Buffer" (44.7 mM $K_2SO_4$, 10 mM $MgSO_4$, 106 mM sucrose, 5 mM glucose, 20 mM Tris-$H_2SO_4$, 3.5 mg/mL BSA, pH to 8.2 with $H_2SO_4$), which preserves the parasites in an intracellular-like state[37]. For "protruded conoid" samples, parasites were kept in HEPES

pH 7.4 buffered saline after release from cells. All parasites were passed through a 5-µm filter to remove cell debris, washed in an appropriate buffer, and collected by centrifugation for 10 min at 300×g. Parasites were resuspended in the respective buffer and incubated for 10 min at 37 °C with vehicle (retracted) or 10 µM calcium ionophore (protruded; A23187; Cayman Chemicals). About 4 µl of the extracellular parasites were pipetted onto a glow-discharged (30 s at −30 mA) copper R2/2 holey carbon grid (Quantifoil Micro Tools GmbH, Jena, Germany). Samples were back-blotted with a Whatman filter paper (grade 1) for 3-4 s to remove excess liquid, then the grid was quickly plunge frozen into liquid ethane using a homemade plunge freezer. For detergent-extracted samples, membranes were extracted by the addition of Triton-X-100 in HBSS for 3–4 min before washing briefly in HBSS and back-blotting, as above. Vitrified grids were mounted in notched Autogrids for cryo-FIB-milling (Thermo Fisher Scientific, MA, USA) and stored in liquid nitrogen until used.

### Cryo-FIB milling
Autogrids with vitrified *N. caninum* cells were loaded into a cryo-shuttle and transferred into an Aquilos dual-beam instrument (FIB/SEM; Thermo Fisher Scientific) equipped with a cryo-stage that is pre-cooled to minus 185 °C. Tile-set images of the grid were generated in SEM mode, and the cells suitable for cryo-FIB milling were targeted using the Maps software (Thermo Fisher Scientific). To protect the specimen and enhance conductivity, the sample surface was sputter-coated with platinum for 20 s at minus 30 mA current and then coated with a layer of organometallic platinum using the gas injection system pre-heated to 27 °C for 5 s at a distance of 1 mm before milling[67,68]. Bulk milling was performed with a 30 kV gallium ion beam of 50 pA perpendicular to the grid on two sides of a targeted cell. Then, the stage was tilted to 10°–18° between the EM grid and the gallium ion beam for lamella milling. For rough milling, the cell was milled with 30 kV gallium ion beams of 30 pA current, followed by 10 pA for polishing until the final lamella was 150–200 nm thick. The milling process was monitored by SEM imaging at 3 keV and 25 pA. A total of 178 lamellae of *N. caninum* were milled over multiple sessions.

### Cryo-ET imaging
Cryo-FIB-milled lamellae of vitrified apical regions of the parasites were imaged using a 300 keV Titan Krios transmission electron microscope (Thermo Fisher Scientific) equipped with a Bioquantum post-column energy filter (Gatan, Pleasanton, CA) used in zero-loss mode with a 20 eV slit width and a Volta Phase Plate with −0.5 µm defocus[69]. The microscope control software SerialEM v4.0.8 was utilized to operate the Krios and collect tilt series from 56° to −56° in 2° increments using a dose-symmetric tilting scheme in the lose-dose mode[70,71]. Images were captured using a 5k × 6k K3 direct electron detection camera (Gatan) at a magnification of 26,000x (3.15 Å pixel size). The counting mode of the K3 camera was used, and for each tilt image, 15 frames (0.04 s exposure time per frame, the dose rate of ~28 e/pixel/s; frames were recorded in super-res mode and then binned by 2) were captured. The total electron dose per tilt series was limited to 100 e/Å$^2$. In total, 167 tilt series were collected from the cryo-FIB-milled lamella of native parasites, but only about 10% of them contained the full or partial apical complex. 19 tilt series were recorded of the apical region of detergent-treated (not cryo-FIB milled) parasites.

There are several factors that contribute to the low yield of the apical complex in our experiments. Firstly, we have plunge-frozen the intact parasites in a relatively thick (>1 µm) layer of ice. It is challenging to determine the apical vs. basal ends of the ice-embedded cells in the cryo-FIB milling instrument. Secondly, during the cryo-FIB milling step, the ice thickness was reduced from more than 1 µm to 150–200 nm, removing more than 80% of the volume. Thus, the probability of placing the lamella exactly in the region of the about 300 nm wide conoid is relatively low compared to accidentally milling the apical complex

away during the thinning step. Finally, about 10–15% of lamellae were damaged or surface-contaminated during the transfer step from the milling instrument to the TEM. Future application of fluorescence-guided cryo-FIB milling and autoloader systems for more direct transfer of FIB-milled samples into the TEM could address these issues and increase experiment throughput.

## Data processing and figure generation

The frames of each tilt series image were motion-corrected using MotionCor2 v1.2.3 and then merged using the script extracted from the IMOD v4.9.3 software package[72] to generate the final tilt serial data set. Tilt-series images were aligned either fiducial-less using patch tracking (800 × 800-pixel size) or using dark features as fiducials (e.g., granules from the sputter coat or embedded Gallium from the milling process) using the IMOD software package. Tomographic reconstructions were calculated using both weighted back-projection before subtomogram averaging, and simultaneous iterative reconstruction technique for visualizing raw tomogram data with higher contrast e.g., for particle picking. Of the recorded 167 tilt series of cryo-FIB-milled native parasites, 125 tilt series were reconstructed for further inspection, and 20 of the reconstructed tomograms contained the apical complex (13 in the protruded and seven in the retracted state). Subtomograms that contain the conoid fiber, the ICMT or PCR repeats were extracted from the raw tomograms, aligned, and averaged with missing-wedge compensation using the PEET v1.10.0 program[73,74]. About 1160 and 721 8-nm conoid fiber repeats were selected from the protruded and retracted native conoid tomograms, respectively, and 386 8-nm ICMT repeats were picked from the reconstructed tomograms of detergent-extracted samples. For the PCR average, 180 subtomograms from four tomograms (three protruded and one retracted conoid) were selected, and initial motive lists with starting orientations were generated using the spikeInit functions in IMOD. Fourier shell correlations were calculated using the calcFSC function and plotted using the plotFSC function in IMOD. For the microneme tip, 25 subtomograms from three tomograms (two protruded and one retracted) were extracted, aligned, and averaged. Parameters regarding data collection and processing are summarized in Supplementary Table 1. For visualization of raw tomographic slices, tomograms were denoised for improved clarity using either non-linear anisotropic diffusion or a weighted median filter (smooth filter) implemented in IMOD. Isosurface renderings and cellular segmentation with manual coloring were generated using the UCSF Chimera v1.10.2 software package[75], which was developed by the Resource for Biocomputing, Visualization, and Informatics at the University of California, San Francisco, with support from NIH P41-GM103311. The movie was rendered in Chimera and compressed with ffmpeg v5.1.1.

## Segmentation of the secretory organelles

Rhoptries, micronemes, and vesicles could be distinguished based on their distinctive morphologies. Rhoptries have a unique club shape and could be divided into two separate structural regions—an anterior tubular neck and a posterior bulb located deeper within the cell body. Micronemes are medium-sized, rod-like organelles, and their interior displayed a darker electron density than other organelles. The vesicles that are closely associated with the ICMT or the apical plasma membrane are small (diameter <50 nm), round, or oval-shaped, and their content appeared brighter (i.e., lower electron scattering properties) than the cytoplasm.

## Measurements

In order to measure the conoid dimensions, tomograms were rotated in the IMOD slicer windows, with the bottom plane oriented horizontally. The maximum distance in the longitudinal section across the center of the conoid was then used to determine the conoid's dimensions, as displayed in Supplementary Fig. 5a. Three protruded and three retracted conoids were measured and compared. The lengths of the conoid fibers were measured as follows: After the conoid fiber was manually tracked using the IMOD program, the length of the conoid fibers and the spacing between adjacent conoid fibers were determined using the IMOD commands imodinfo and mtk, respectively. In total, 20 full-length conoid fibers from the protruded conoids and 12 from the retracted conoids were measured. To compare the APRs and PCRs in parasites with protruded or retracted conoid, we measured their diameters as follows: The tomograms were rotated in the IMOD slicer window until cross-sectional views of the PCRs and APRs could be saved (as seen in Supplementary Fig. 5b, c); then the diameters of the rings were determined using the ImageJ (Fiji distribution v1.53c) circle tools. Measurements were made of three APRs and three PCRs from protruded conoids, and three APRs and two PCRs from retracted conoids. The distance between the conoid and the IMC were measured as follows: we positioned the center of the view in the IMOD slicer window at the apical end of the IMC and then rotated the tomogram around this center point in 3D to find the shortest distance between the conoid and the apical end of the IMC.

## Statistics and reproducibility

We have summarized the number of representative tomograms for each figure panel in Supplementary Table S2.

## Reporting summary

Further information on research design is available in the Nature Portfolio Reporting Summary linked to this article.

## Data availability

The subtomogram average density maps of the *Neospora caninum* conoid fiber in the protruded and retracted states, and a global average combining data from both states have been deposited in the Electron Microscopy Data Bank (EMDB) with accession code EMD-28247, EMD-28249, and EMD-26873, respectively. The subtomogram average density maps of the PCRs P1-P2, P3, and the composite map of P1-P2-P3 have been deposited in the Electron Microscopy Data Bank (EMDB) with accession code EMD-28231, EMD-28234, and EMD-28246, respectively. All other data needed to evaluate the conclusions in the paper are present in the paper and/or the Supplementary Materials. Source data are provided with this paper.

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

## Acknowledgements

We thank Daniel Stoddard for training and management of the cryo-electron microscopy facility at the University of Texas, Southwestern Medical Center, which is supported in part by the CPRIT Core Facility Support Award RP170644. This research was supported in part by the computational resources provided by the BioHPC supercomputing facility located in the Lyda Hill Department of Bioinformatics, UT Southwestern Medical Center.

## Author contributions

M.L.R. and D.N. conceived the project; L.G. performed tomogram reconstruction, subtomogram averaging, data analysis, figure, and movie preparation; W.J.O. performed cell culture, sample preparation, and data analysis; K.C. performed cryo-preparation, cryo-ET data collection, and initial image processing; E.R. performed cryo-FIB milling; M.L.R. performed data analysis and wrote the manuscript with help from L.G. and D.N.; D.N. performed data analysis and supervised the overall project. This work was supported by the National Institutes of Health (NIH; R01GM083122 to D.N. and R01AI150715 to M.L.R.), the Cancer Prevention and Research Institute of Texas (CPRIT; RR140082 to D.N.), the National Science Foundation (MCB1553334 to M.L.R.), and the Welch Foundation (I-2075-20210327 to M.L.R.).

## Competing interests

The authors declare no competing interests.
