## [Peer Review File · Nature Communications]

REVIEWER COMMENTS

Reviewer #1 (Remarks to the Author):

Summary:

This manuscript uses cellular cryo-electron tomography (cryo-ET) imaging to visualize and characterize the three-dimensional architecture of the apical complex in apicomplexan parasites. As rationale for pursuing a new sample preparation and imaging approach, the authors show that previous strategies were plagued by flattening artifacts and structural distortions caused by detergent extraction. The authors optimized vitrification conditions and determined that freezing parasites intact in a relatively thick layer of ice followed by thinning by cryo-FIB milling preserved the native architecture of these structures. The authors used pharmacological strategies to capture parasites with conoids in either retracted or protruded states, and revealed incredibly detailed tomographic reconstructions of the architecture of the protruded apicomplexan invasion machinery.

The authors first focus their structural characterization on the pre-conical rings (PCRs), which are the cytoskeletal structures closest to the apical plasma membrane. They use subtomogram averaging to improve the clarity and resolution of the endogenous PCR structures, and reveal a new, previously uncharacterized, smaller PCR ring that sits apical on top of the apical-polar-ring (APR). The authors also identified a ring of "amorphous APR-associated density" (AAD) located between the inner membrane complex (IMC) and the tips of the subpellicular microtubules, which has historically been unresolved in previous conventional EM studies, although recent cryo-ET reports a similar structure as well. Next, the authors focus on the overall conoid structure itself, and demonstrate that the overall shape and architecture of the conoid is similar in both the protruded and retracted states. The authors make an observation that the conoid appears to be "tilted" in their tomograms, suggesting that protrusion may proceed through a "wobbling" of the rigid conoid on top of the APR. The authors use subtomogram averaging to reveal more detailed structures of the "c-shape" architecture of the nine tubulin protofilaments that make up the conoid fibers. Their structures reveal densities on both the "inner" and "outer" sides of the tubulin protofilaments, which they predict to be microtubule associated proteins (MAPs) although the precise identity of these densities is unknown. Lastly, the authors characterize the ultrastructure of the vesicles surrounding the subpellicular microtubules, and demonstrate that these secretory vesicles appear to be segregated within the conoid complex, with regularly-spaced, connected smaller vesicles one side, and the micronemes and rhoptries preferentially localized on the opposite side.

Significance:

This manuscript harnesses state-of-the-art advancements in sample preparation procedures (i.e., cryo-focused ion beam milling) the field of cellular cryo-electron tomography to reveal the complex architecture of the invasion machinery of apicomplexan parasites. The authors present a thorough structural analyses of the apical complex during both retraction and protrusion, with a particular focus on the cytoskeletal components and their interactions that mediate host cell invasion. This manuscript is timely with the recent development of the field of "structural parasitology", and is a nice complement to several other manuscripts from other groups using similar approaches to reveal other aspects and machinery involved in parasitic invasion.

Suggested areas of improvement:

Overall, the majority of the conclusions and claims presented in this manuscript are supported by sufficient evidence throughout. However, this manuscript could benefit from additional information and clarification in a few sections, highlighted below:

Overall, there is a notable lack of details regarding the quantitative and statistical analyses performed throughout the manuscript, which makes it difficult to critically evaluate some of the associated claims made based on these quantitative observations (see below). I suggest the authors include details of sample size, statistical test, quantitative analyses, etc. for each relevant figure in the figure legends and/or the methods section.

The authors show that the overall structural features of the conoid (dimensions, pitch, spacing, etc.) appear to be unchanged in both the protruding and retracted apical complex. They also observe an asymmetry in the "docking" of the conoid such that the distances between the basal edge of the conoid and the apical rim of the IMC are distinct from opposing sides. Based on these observations, the authors predict that the conoid may "wobble" as a rigid body, as opposed to the current model which predicted this movement may be "spring-like" during invasion. This is intriguing, however it is difficult to make claims about dynamics such as a "wobbling" motion from static 3D tomographic reconstructions. Furthermore, it is unclear based on the evidence presented how robust this asymmetric "tilting" observation is across all tomograms in their dataset. Could the authors comment on how these measurements were made? What criteria were used to determine which slice the measurements were taken from? Were multiple measurements taken in each tomogram for each dimension and then averaged? Did the authors perform this analysis in 2D by calculating the distance between two points in the tomogram, or were 3D surfaces used for the calculations? Additional details on the methodology in this section would be useful to provide more convincing argument in favor of the asymmetry that leads to the proposed "wobble" model.

It is not obvious from the images and methods presented in this manuscript how the authors were able to unambiguously differentiate between the different membrane-bound compartments (i.e., micronemes, rhoptries, secretory vesicles). Could the authors please comment on how these distinctions were made for their segmentation analyses?

For the filamentous densities in figure 3, although two of them appear very obvious in the image presented, it is extremely difficult to be convinced for some of the others in the tomogram. I suggest the authors present more slices/views to better convince the reader that these are indeed actin-like filaments.

For line 424: "With the ability to capture the cellular dynamics of active secretion, we were able to observe a docked microneme in the process of secretion."; and line 237: Our data unambiguously show that microneme secretion can occur during conoid protrusion, events that have been indirectly correlated by other studies." Proximity does not necessarily equal active secretion, especially given the context of this sample that is lacking the host cell. Could the authors comment and/or justify this claim more clearly? How many examples did they observe of docked (and potentially actively secreting) micronemes in their data?

Line 312: "Notably, we did not observe this bridging density at the apical, membrane-docked vesicle (Figure 7A). Whereas the apical vesicle likely originated from the ICMT-associated vesicle chain, in docking with the rhoptry and plasma membrane, it appears to have been released from the vesicle-chain and its connections." What is the sample size for this observation?

For lines 670: "a helical scaffold (dark rose) associated with the rhoptry membrane". Is it truly helical? (i.e., does the power spectrum of this area show layer lines representing helical geometry?)

The authors describe the organization of the organelles in the apical complex, however it is unclear whether or not this analysis pertains to the retracted or protruded state. Are there differences in the organization of the secretory organelles between these two states?

Line 307: "We observed a single vesicle just apical of the ICMT that appears docked to the density of the recently described "rosette," and which has been associated with facilitating rhoptry secretion (13). Consistent with this function, we observe that the membrane-docked vesicle connects to the apical tip of 1-2 rhoptries." It is unclear which figure this text is referring to, I believe it should be Supp Fig 7J-N?

Overall, there are several instances in which figure panels do not match up to the figure legends, or image labels are not defined in legend:

Figure 1: no B'-D' pseudocolored figure panels; figure panels show A-G, but legend has A-J.
Figure 2: legend is confusing as written.
Figure 4: to what features do the blue arrows in Figure 4A and C correspond? The pseudocolored versions of c' and d' are unlabeled.
Figure 5: mismatch between figure panels and legend.

It would be helpful to include a cartoon version of the parasite in relevant figures to orient the reader to the different components, views, and states of retraction of the apical complex that are discussed throughout the paper.

Reviewer #2 (Remarks to the Author):

Summary: This manuscript describes structural details of the Neospora apical complex revealed by cryo-FIB-milling and cryo-ET. Significant observations in this work include the findings that the conoid itself is a rigid body and that the apical polar ring is not rigid and can change diameter. These conclusions are distinct from previous interpretations in studies of detergent-extracted material.

This manuscript has some spectacular images and important conclusions, but the labeling of these should be modified to improve clarity in the context of both font size and font color. In addition, the legends do not match the figures which are complex and data heavy. Attention to these details is critical correction for the manuscript to be appreciated by a larger scientific community. Lastly, I think it is key to be clear and transparent about which parasites this data is relevant to. That is, the FIB structures were collected for Neospora, a close relative to Toxoplasma. Also, it needs to be obvious that not all components of the Neospora/Toxoplasma apical complex are found in Plasmodium.

Detailed corrections/revisions:

Line 23: "The apical complex is a conserved cytoskeletal structure that organizes the secretory and invasion machinery of all apicomplexan parasites, which cause, e.g., malaria and toxoplasmosis." This sentence seems to be missing something and it doesn't encompass the rhoptries and micronemes. How about: "The apical complex is a specialized collection of secretory and cytoskeletal machinery in apicomplexan parasites, which include human pathogens such as the agents of malaria and toxoplasmosis."

Line 46: "In Apicomplexa these structures are called..." should be "the Apicomplexa" or apicomplexans

Line 52: "The apical complex itself is a ~250 nm long structure..." The apical complex is defined as the set of cytoskeletal AND secretory organelles (micronemes and rhoptries). This should be stated, the cytoskeletal elements in it could be referred to as the conoid complex or the apical complex cytoskeleton.

Line 132: "they have previously been analyzed in detail (38)." You should also cite reference 35 here: "Cryo-ET of Toxoplasma parasites gives subnanometer insight into tubulin-based structures" (Boothroyd)

Line 215: "As the conoid protrudes, the relative position of the APR shifts from level with the PCRs to basal to the conoid (compare Figure 4A,C,C' with 4B,D,D')." To readers outside of the field this may appear confusing because it doesn't reflect the frame of reference of the parasite body. How about "While the PCRs are level with the APR when the conoid is retracted, after extension, the conoid base is located in the region of the APR." I don't know that this is better but maybe it helps you think of ways to clarify this point.

Line 294: This analysis demonstrates that the ICMT, unlike the conoid fibers, have their minus-ends oriented apically. This observation suggests that the ICMT, unlike other

apicomplexan MT structures, exhibits dynamic instability at its plus-end (Figure 6G), and may explain why the observed ICMT lengths vary largely between samples. I'm a bit confused about the argument here. Many microtubules have their minus ends capped, with non-dynamic and dynamic examples. While dynamic instability occurs in many settings, I'm not sure that capping is an argument for this behavior. Is your argument that the different observed lengths of the microtubules evidence of dynamic instability? Also, do you have evidence that both microtubules are organized with the same polarity? (I think so, but you may want to explicitly state it of so. Do you want to say something about motors for trafficking based on microtubule polarity? (I see you have some thoughts on this in the discussion, but nothing about the ICMT polarity with motors).

Line 403: "Thus, the ICMT may have been conserved in the ancestral species..." How about Thus, the ICMT was likely present in the common ancestral species..." (Conservation is forwards not backwards)

Line 441: "These significant advances allowed us to interrogate the native structure of the apical complex and its movements during retraction and protrusion." Specify the conoid: "These significant advances allowed us to interrogate the native structure of the conoid complex and its movements during retraction and protrusion.

Line 564: "views, apical is oriented towards the top" This is missing a few words: "views, the apical tip is oriented towards the top"

Line 571: The figure legend for Figure 1 (panels A-G) must also incorporate descriptions from Figure 2 which has panels H-J. This is extremely important confusing because J must be what is labeled F.

Line 598: "shows filamentous densities (magenta) connect the conoid (orange)" – are these the densities that you suggest are F-actin-like? If so, it would be helpful to state that here.

Figure 1:

Panel A: I strongly suggest making the labels a larger font size and perhaps not bold. The orange conoid filament color is particularly difficult to see (maybe use a lighter shade?), as is the black and white PM label. "Other labels and coloring see below" Please describe what the purple arrows are in the first occurrence (panel a) so the reader can evaluate this panel effectively.

Panel C: Please use a different color/shade to number the conoid filaments: the orange is difficult to see, even on a computer screen. If every third filament was numbered, the font size could be larger.

Reviewer #3 (Remarks to the Author):

The authors uncover multiple new interesting ultrastructures of the invasion machinery of the apicomplexan parasites. This is done by cryo-FIB milling and cryo-ET of the parasite. They provide new insights into the conoid fiber structure by sub-tomogram averaging in the protruded and the retracted states and challenge the conventional spring model. They also observe for the first time multiple new "layers" of this giant complex, including a new layer of the pre-conoidal rings that they characterize by sub-tomogram averaging. The protruded and retracted states are prepared using relevant buffers, and the parasites are not actively engaged in the act of invasion. The data quality is superb and the amount of data analyzed is significant considering the challenges of in situ cryo-ET. This manuscript will open new avenues of research, including the identification of new components of this complex, and uncovering the mechanistic details of the apical complex during invasion.

Major comments:

1. The panels of figure 1 and the legend of figure 1 do not match. The panels go from A to G but the legends go from A to J. Also the descriptions do not match the panels.
2. The quality of the tomograms is good, and it seems like most of the data is segmented. I suggest that the authors use these segmentations to present their results in 3D representations rather than 2D tomographic slices. That is true for most of the figures, specially the main figures. The audience is interested in the structures and the relationship between different components, and that is difficult to grasp from 2D slices of busy-looking tomograms, although I appreciate the effort that the authors put in to show the kind of data that their observations are based on. I suggest that the authors find a balance between showing tomogram slices and 3D structures in the figures, mainly by reducing the number of panels with 2D slices. For example, panel F of figure 1 should be the first panel of that figure with the structures annotated on the segmentation itself. That is the most interesting result and it should be amplified.
3. I found the organization of the figures very confusing. For example, for most of the observations in the results section, there is usually more than one figure referenced, sometimes two main figures and a supplementary figure. But looking at those figures doesn't clearly get the author's observation across. The reader must do a lot of mental gymnastics to picture what is going on and has to put in a lot of effort to put the pieces together, again mainly because the figures are busy tomogram slices with annotations that are found in the legend or elsewhere. I generally found the figures to be very ineffective in getting the message across. I think each figure should have a clear message that is immediately obvious to the reader. Another example is that I was surprised that the first figure does not have a side-by-side comparison of the retracted and the protruded state using segmentations. Another example is that the different components of this giant complex are introduced on a busy tomogram slice in figure 1A, instead of a simple schematic or on figure 1F.
4. A supplemental figure describing the genetic tree of the different species discussed in the manuscript would be helpful. Otherwise, the conclusions regarding the differences between the apical complex of different species should be made more clear.
5. The introduction starts with a paragraph about the apical complex, it's followed by a paragraph about the genetic tree and the micronemes and rhoptries, and then the next paragraph goes back to discussing the apical complex. I think the flow would be better if the third paragraph switched places with the second paragraph.
6. The paragraph describing in situ cryo-ET that starts at line 65 is lengthy and I'm not sure that there is a need for such a long introduction to the technique. I also found the bottom-up and top-down terminology unnecessary and unclear. Maybe it would suffice to say that in situ cryo-ET provides context to the structural analysis and that it might reveal structures that were not seen before in reconstituted studies.
7. The choice of *Toxoplasma gondii* and *Neospora caninum* for the in situ cryo-ET analysis is not clear (Line 77). why these two specifically?
8. Line 80: "without the compression and deformation that plague unmilled samples". This phrasing is misleading because not all unmilled samples are compressed.
9. The title of the first result "Cryo-FIB-milling of intact parasites..." is very generic and sounds like what one would find in a methods paper. I think the first result should be a clear statement about the observed structures, and not a description of in situ cryo-ET. The content of this section is also focused on methods. I think it's clear to the community what each of these methods provide.
10. Line 114: 10 micro molar?
11. Line 204: the reasoning sounds circular. It sounds like the authors are just describing what they mean by conformational change in the context of the retracted and protruded states. I think the opening sentence should state that the authors set out to investigate the spring model by looking at gross changes in the dimensions.
12. The conclusion that the spring model is incorrect and the introduction of the rigid body model are not corroborated enough. These states were induced chemically, and the organism is not actively invading a host, so one could argue that the protruded state might be an intermediate/alternative state and that's why the spring-like motion is absent. The authors should rule out these possibilities in clear statements. The authors rather argue

that in situ cryo-ET provides close-to-native conditions (which is true) therefore, the induced state is exactly the protruded state.

13. Line 240: at 3 nm resolution it is a big statement to say that positioning of the protofilaments is indistinguishable between the two states. Figure 5E shows differences between the two states and the docking in figure 5C and 5D does not seem to be tight.

14. Line 257: The increase in resolution is not interpretable because merely the higher number of particles contributing to the map could inflate the estimated resolution. It certainly cannot be used to prove the rigidity of the structure, specially since it's not clear how much difference is a meaningful difference in this context. I think this data serves as a lack of evidence for severe compression or expansion, but It does not prove that the two states have identical structures.

15. There is a lack of referencing figures in the discussion.

16. The authors milled 178 lamella, collected 167 tomograms, of which 20 had the apical complex. The authors should comment on the nature of the uncertainties that limited the throughput during each step (milling and data collection) in the methods section.

17. I think there are two story lines in this paper that do not necessarily go well together. One is focused on methods and comparing methods in detail. And one is focused on structure and mechanism. I think the flow would be much better and the manuscript would be much easier to read and the results would be more impactful if the authors stick to one story line, that is the structural analysis.

Please find below our point-by-point answers to the reviewers' comments & suggestion:

The reviewers' comments are copied below (*italic black/gray*) and our point-by-point answers are in **blue font** (text cited from the manuscript is in *red italic font*). Where appropriate, we refer to specific sections in the revised manuscript where changes have been made. We have tracked changes in the manuscript, but please note that we did not track moved paragraphs to allow us to highlight changes within those paragraphs. Page numbers indicated in this document refer to the "tracked-changes" version and may not match the pdf version where changes are left untracked..

REVIEWER COMMENTS

Reviewer #1 (Remarks to the Author):

Summary:

This manuscript uses cellular cryo-electron tomography (cryo-ET) imaging to visualize and characterize the three-dimensional architecture of the apical complex in apicomplexan parasites. As rationale for pursuing a new sample preparation and imaging approach, the authors show that previous strategies were plagued by flattening artifacts and structural distortions caused by detergent extraction. The authors optimized vitrification conditions and determined that freezing parasites intact in a relatively thick layer of ice followed by thinning by cryo-FIB milling preserved the native architecture of these structures. The authors used pharmacological strategies to capture parasites with conoids in either retracted or protruded states, and revealed incredibly detailed tomographic reconstructions of the architecture of the protruded apicomplexan invasion machinery.

The authors first focus their structural characterization on the pre-conical rings (PCRs), which are the cytoskeletal structures closest to the apical plasma membrane. They use subtomogram averaging to improve the clarity and resolution of the endogenous PCR structures, and reveal a new, previously uncharacterized, smaller PCR ring that sits apical on top of the apical-polar-ring (APR). The authors also identified a ring of "amorphous APR-associated density" (AAD) located between the inner membrane complex (IMC) and the tips of the subpellicular microtubules, which has historically been unresolved in previous conventional EM studies, although recent cryo-ET reports a similar structure as well. Next, the authors focus on the overall conoid structure itself, and demonstrate that the overall shape and architecture of the conoid is similar in both the protruded and retracted states. The authors make an observation that the conoid appears to be "tilted" in their tomograms, suggesting that protrusion may proceed through a "wobbling" of the rigid conoid on top of the APR. The authors use subtomogram averaging to reveal more detailed structures of the "c-shape" architecture of the nine tubulin protofilaments that make up the conoid fibers. Their structures reveal densities on both the "inner" and "outer" sides of the tubulin protofilaments, which they predict to be microtubule associated proteins (MAPs) although the precise identity

of these densities is unknown. Lastly, the authors characterize the ultrastructure of the vesicles surrounding the subpellicular microtubules, and demonstrate that these secretory vesicles appear to be segregated within the conoid complex, with regularly-spaced, connected smaller vesicles one side, and the micronemes and rhoptries preferentially localized on the opposite side.

Significance:

This manuscript harnesses state-of-the-art advancements in sample preparation procedures (i.e., cryo-focused ion beam milling) the field of cellular cryo-electron tomography to reveal the complex architecture of the invasion machinery of apicomplexan parasites. The authors present a thorough structural analyses of the apical complex during both retraction and protrusion, with a particular focus on the cytoskeletal components and their interactions that mediate host cell invasion. This manuscript is timely with the recent development of the field of “structural parasitology”, and is a nice complement to several other manuscripts from other groups using similar approaches to reveal other aspects and machinery involved in parasitic invasion.

We thank the reviewer for the positive evaluation.

Suggested areas of improvement:

Overall, the majority of the conclusions and claims presented in this manuscript are supported by sufficient evidence throughout. However, this manuscript could benefit from additional information and clarification in a few sections, highlighted below:

1) Overall, there is a notable lack of details regarding the quantitative and statistical analyses performed throughout the manuscript, which makes it difficult to critically evaluate some of the associated claims made based on these quantitative observations (see below). I suggest the authors include details of sample size, statistical test, quantitative analyses, etc. for each relevant figure in the figure legends and/or the methods section.

We thank the reviewer for this suggestion and have included additional details about quantitative and statistical analyses where relevant. For details, please see below.

2) The authors show that the overall structural features of the conoid (dimensions, pitch, spacing, etc.) appear to be unchanged in both the protruding and retracted apical complex. They also observe an asymmetry in the “docking” of the conoid such that the distances between the basal edge of the conoid and the apical rim of the IMC are distinct from opposing sides. Based on these observations, the authors predict that the conoid may “wobble” as a rigid body, as opposed to the current model which predicted this movement may be “spring-

like” during invasion. This is intriguing, however it is difficult to make claims about dynamics such as a “wobbling” motion from static 3D tomographic reconstructions. Furthermore, it is unclear based on the evidence presented how robust this asymmetric “tilting” observation is across all tomograms in their dataset. Could the authors comment on how these measurements were made? What criteria were used to determine which slice the measurements were taken from? Were multiple measurements taken in each tomogram for each dimension and then averaged? Did the authors perform this analysis in 2D by calculating the distance between two points in the tomogram, or were 3D surfaces used for the calculations? Additional details on the methodology in this section would be useful to provide more convincing argument in favor of the asymmetry that leads to the proposed “wobble” model.

We agree with the reviewer that cryo-EM/ET cannot visualize **dynamics**, because the samples are frozen. Instead, cryo-EM/ET provides structural snapshots, including structural conformations at a given functional state; e.g. our data show that the conoid is present in the same structural conformation in both functional states, the protruded or retracted, supporting a “rigid body” translocation mechanism (instead of the current “spring like” model). However, the word “wobble” implies dynamics that we cannot assess given our current data - which we did not intend – therefore we have exchanged “wobble” for a detailed description of the observed relationship between the conoid and surrounding structures. In addition, we have added a paragraph (called “**Measurements**”) to the Methods section, describing in detail how all measurements were made (e.g. distances and dimensions were measured in 3D using IMOD).

3) It is not obvious from the images and methods presented in this manuscript how the authors were able to unambiguously differentiate between the different membrane-bound compartments (i.e., micronemes, rhodoptries, secretory vesicles). Could the authors please comment on how these distinctions were made for their segmentation analyses?

We thank the reviewer for the suggestion, and we have now added a paragraph (called “**Segmentation of the secretory organelles**”) to the Methods section, describing in detail how different cellular structures and compartments were identified. Briefly, the three organelles have distinct locations, shapes, dimensions, and electron densities of their contents, allowing unambiguous differentiation.

4) For the filamentous densities in figure 3, although two of them appear very obvious in the image presented, it is extremely difficult to be convinced for some of the others in the tomogram. I suggest the authors present more slices/views to better convince the reader that these are indeed actin-like filaments.

We agree with the reviewer; because the filaments are distributed in 3D, it is not possible to visualize several of them well-oriented in a single 2D tomographic slice. As per the reviewer's suggestion, we have modified Figure 2 to now include a 3D rendering (Fig. 2G);

in addition, we now show for the overview slice in Fig. S4 also different tomographic slices that are optimized to visualize individual filament (Fig. S4I-L).

For line 424: “With the ability to capture the cellular dynamics of active secretion, we were able to observe a docked microneme in the process of secretion.”; and line 237: Our data unambiguously show that microneme secretion can occur during conoid protrusion, events that have been indirectly correlated by other studies.” Proximity does not necessarily equal active secretion, especially given the context of this sample that is lacking the host cell. Could the authors comment and/or justify this claim more clearly? How many examples did they observe of docked (and potentially actively secreting) micronemes in their data?

We would like to clarify the physiological relevance of microneme secretion in extracellular parasites. In addition to microneme secretion of the invasion co-receptor (AMA1; required for exiting cells), micronemes also secrete adhesins that are absolutely required for initiating movement from the parasite vacuole and for moving to locate a new host cell (before invasion or contact with host cells). Thus the lack of a host cell is immaterial for microneme secretion. We have attempted to clarify this point in the introduction and results, respectively, with the following sentences: “*Whereas the rhoptry secretion requires close contact with the host cell plasma membrane, micronemes are thought to secrete continuously while the parasites are extracellular*” (page 4), and: “*Whereas micronemes secrete continuously to promote the motility of extracellular parasites, they are also intimately involved in triggering host cell invasion*” (page 17).

We agree with the reviewer, that “proximity” does not necessarily mean “interaction”. However, in the here observed case, there is clearly interaction (i.e. electron density) between the microneme membrane and plasma membrane. This said, we have only observed this event once and thus we have toned down our interpretation in the discussion (pages 19-20).

Line 312: “Notably, we did not observe this bridging density at the apical, membrane-docked vesicle (Figure 7A). Whereas the apical vesicle likely originated from the ICMT-associated vesicle chain, in docking with the rhoptry and plasma membrane, it appears to have been released from the vesicle-chain and its connections.” What is the sample size for this observation?

Of the 20 tomograms that contained at least parts of the conoid complex, 6 tomograms (5 protruded, 1 retracted) contained the entire chain of vesicles and the apical vesicle in longitudinal orientation (which is the ideal orientation for visualizing linker densities between the vesicles, as compared to the 4 tomograms where the conoid is in a cross-sectional orientation). In the remaining tomograms the apical vesicle was milled away by cryo-FIB. In those 6 tomograms with the entire vesicle chain/apical vesicle, we always

observed bridging densities between the vesicles along the ICMT, but never with the apical-most vesicle. We have added this quantification to the text: *“(Figure 7A; n=6 tomograms with the vesicle chain and apical vesicle present; note that in other tomograms the apical vesicle was milled away by cryo-FIB).”*

For lines 670: “a helical scaffold (dark rose) associated with the rhoptry membrane”. Is it truly helical? (i.e., does the power spectrum of this area show layer lines representing helical geometry?)

In a previous study this scaffold was averaged using helical fitting (Mageswaran, et al 2021; ref 51). However, in our data the power spectrum of the rhoptry necks do not show layer lines consistent with helical geometry. This could be due to the low signal-to-noise ratio and limited information (missing wedge) in the raw tomograms, or because the pitch is somewhat variable (heterogeneity). Because of this ambiguity, we have changed our language in the figure legend and text to: *“a spiraling scaffold (dark rose) associated with the rhoptry membrane.”* (page 40), *“Within the apical region of the parasite, a spiral-shaped density corkscrews along the rhoptry membranes”* (page 15), and *“we and others identified a density spiraling around the rhoptries [...] These spiraling filaments may act...”* (page 20).

The authors describe the organization of the organelles in the apical complex, however it is unclear whether or not this analyses pertains to the retracted or protruded state. Are there differences in the organization of the secretory organelles between these two states?

We thank the reviewer for pointing out this oversight. We did not observe any differences in the secretory organelles and their arrangements between the two states. To clarify this for the readers, we have now a) include images of a fully segmented tomogram from a retracted parasite (Fig. 6B,E,G), and b) modified the text to read: *“we observe that distinct secretory organelles (the rhoptries and micronemes) segregate to opposite sides of the ICMT in tomograms from both protruded and retracted conoids (Figure 6)”*

Line 307: “We observed a single vesicle just apical of the ICMT that appears docked to the density of the recently described “rosette,” and which has been associated with facilitating rhoptry secretion (13). Consistent with this function, we observe that the membrane-docked vesicle connects to the apical tip of 1-2 rhoptries.” It is unclear which figure this text is referring to, I believe it should be Supp Fig 7J-N?

We apologize for missing this figure reference. We have added the appropriate figure references in the revised manuscript (Fig. 7B-B'; and Supplemental Fig. S8A); please note

that figure panel S8A has been newly added to show an apical vesicle connected to 2 rhoptries.

Overall, there are several instances in which figure panels do not match up to the figure legends, or image labels are not defined in legend:

Figure 1: no B'-D' pseudocolored figure panels; figure panels show A-G, but legend has A-J.

Figure 2: legend is confusing as written.

Figure 4: to what features do the blue arrows in Figure 4A and C correspond? The pseudocolored versions of c' and d' are unlabeled.

Figure 5: mismatch between figure panels and legend.

We sincerely apologize for all issues with the figure legends, and have updated the text and figure legends appropriately.

It would be helpful to include a cartoon version of the parasite in relevant figures to orient the reader to the different components, views, and states of retraction of the apical complex that are discussed throughout the paper.

We thank the reviewer for this excellent suggestion, and we have now added a cartoon overview of the parasite in Figures 1A, which we reuse in Figures 2A, 5A, and Suppl. Fig. S7A, to orient the reader to the relevant structures.

Reviewer #2 (Remarks to the Author):

Summary: This manuscript describes structural details of the Neospora apical complex revealed by cryo-FIB-milling and cryo-ET. Significant observations in this work include the findings that the conoid itself is a rigid body and that the apical polar ring is not rigid and can change diameter. These conclusions are distinct from previous interpretations in studies of detergent-extracted material.

This manuscript has some spectacular images and important conclusions, but the labeling of these should be modified to improve clarity in the context of both font size and font color.

We thank the reviewer for the very positive evaluation. We have modified the labeling as suggested (for details please see below).

In addition, the legends do not match the figures which are complex and data heavy. Attention to these details is critical correction for the manuscript to be appreciated by a larger scientific community.

We agree with the reviewer and sincerely apologize for all issues with the figure legends. We have updated the figure legends and figure references in the text appropriately.

Lastly, I think it is key to be clear and transparent about which parasites this data is relevant to. That is, the FIB structures were collected for Neospora, a close relative to Toxoplasma. Also, it needs to be obvious that not all components of the Neospora/Toxoplasma apical complex are found in Plasmodium.

Please see details below how we have addressed these points.

Detailed corrections/revisions:

1) Line 23: *“The apical complex is a conserved cytoskeletal structure that organizes the secretory and invasion machinery of all apicomplexan parasites, which cause, e.g., malaria and toxoplasmosis.” This sentence seems to be missing something and it doesn’t encompass the rhoptries and micronemes. How about: “The apical complex is a specialized collection of secretory and cytoskeletal machinery in apicomplexan parasites, which include human pathogens such as the agents of malaria and toxoplasmosis.”*

We thank the reviewer for the excellent suggestion. We have changed the sentence to *“The apical complex is a specialized collection of cytoskeletal and secretory machinery in apicomplexan parasites, which include the pathogens that cause malaria and toxoplasmosis.”* to ensure we meet the word limits for the abstract.

2) Line 46: *“In Apicomplexa these structures are called...”* should be *“the Apicomplexa”* or *apicomplexans*

Thanks, this was changed to *“In apicomplexans...”*

3) Line 52: *“The apical complex itself is a ~250 nm long structure...”* The apical complex is defined as the set of cytoskeletal AND secretory organelles (micronemes and rhoptries). This should be stated, the cytoskeletal elements in it could be referred to as the conoid complex or the apical complex cytoskeleton.

Thanks, this was changed to *“The apical complex cytoskeleton itself is a ...”*

4) Line 132: *“they have previously been analyzed in detail (38).”* You should also cite reference 35 here: *“Cryo-ET of Toxoplasma parasites gives subnanometer insight into tubulin-based structures”* (Boothroyd)

We thank the reviewer for the suggestion and now also cite reference 35 at this point.

5) Line 215: *“As the conoid protrudes, the relative position of the APR shifts from level with the PCRs to basal to the conoid (compare Figure 4A,C,C’ with 4B,D,D’).”* To readers outside of the field this may appear confusing because it doesn’t reflect the frame of reference of the parasite body. How about *“While the PCRs are level with the APR when the conoid is retracted, after extension, the conoid base is located in the region of the APR.”* I don’t know that this is better but maybe it helps you think of ways to clarify this point.

We thank the reviewer for their help with this difficult wording; we struggled in writing this to make it clear. We have changed it to *“Whereas the apical tip of the conoid and the PCRs are level with the APR when the conoid is retracted, after extension of the conoid, the base of the conoid is located in the region of the APR.”* We hope the reviewer agrees this is much improved.

6) Line 294: *This analysis demonstrates that the ICMT, unlike the conoid fibers, have their minus-ends oriented apically. This observation suggests that the ICMT, unlike other apicomplexan MT structures, exhibits dynamic instability at its plus-end (Figure 6G), and may explain why the observed ICMT lengths vary largely between samples. I’m a bit confused about the argument here. Many microtubules have their minus ends capped, with non-dynamic and dynamic examples. While dynamic instability occurs in many settings, I’m not sure that capping is an argument for this behavior. Is your argument that the different observed lengths of the microtubules evidence of dynamic instability? Also, do you have evidence that both microtubules are organized with the same polarity? (I think so, but you may want to explicitly state it of so. Do you want to say something about motors for trafficking based on microtubule polarity? (I see you have some thoughts on this in the discussion, but nothing about the ICMT polarity with motors).*

The reviewer makes a good point. We now explicitly state that our analysis demonstrates that both microtubules in the ICMT pair are oriented in the same direction (minus-end apical). We apologize that we made a bit of a logical jump and left out one piece of reasoning. Our reasoning for dynamic instability is two-fold: (a) The plus-end appears to be uncapped and in a splayed conformation, which is typical of a “dynamic” (i.e. growing or shrinking) microtubule. Therefore, we have added the following sentence: *“Moreover, the basal, plus-end of the ICMT pair exhibits the splayed morphology typical of a dynamic microtubule (Figure 7H).”* In addition, (b) The ICMT are of varying lengths, and – in contrast to subpellicular microtubules or conoid filaments - appear to be partially unstable upon detergent extraction; thus, previously reported measurements of the ICMT length from extracted samples have been both varied and consistently shorter than those that we measure from intact native parasites. Therefore, we modified the text to read *“... may*

explain why the previously reported ICMT lengths vary largely between samples and are consistently shorter in extracted samples versus those that we measure from intact, native parasites.”

Line 403: *“Thus, the ICMT may have been conserved in the ancestral species...” How about Thus, the ICMT was likely present in the common ancestral species...” (Conservation is forwards not backwards)*

We have changed the text according to the reviewer's suggestion. *“Thus, the ICMT were likely present ...”*

Line 441: *“These significant advances allowed us to interrogate the native structure of the apical complex and its movements during retraction and protrusion.” Specify the conoid: “These significant advances allowed us to interrogate the native structure of the conoid complex and its movements during retraction and protrusion.*

We have changed the text according to the reviewer's suggestion. *“...the conoid complex and its movements...”*

Line 564: *“views, apical is oriented towards the top” This is missing a few words: “views, the apical tip is oriented towards the top”*

We have changed the text according to the reviewer's comment (and improved the entire Figure legend).

Line 571: *The figure legend for Figure 1 (panels A-G) must also incorporate descriptions from Figure 2 which has panels H-J. This is extremely important confusing because J must be what is labeled F.*

We sincerely apologize for all issues with the figure legends, which happened during editing and moving files between authors. We have updated the figure legend to appropriately describe the figures.

Line 598: *“shows filamentous densities (magenta) connect the conoid (orange)” – are these the densities that you suggest are F-actin-like? If so, it would be helpful to state that here.*

We have changed the text according to the reviewer's suggestion to *“shows filamentous actin-like densities”.*

Figure 1:

Panel A: *I strongly suggest making the labels a larger font size and perhaps not bold. The orange conoid filament color is particularly difficult to see (maybe use a lighter shade?), as is the black and white PM label. “Other labels and coloring see below” Please describe what the*

purple arrows are in the first occurrence (panel a) so the reader can evaluate this panel effectively.

Panel C: Please use a different color/shade to number the conoid filaments: the orange is difficult to see, even on a computer screen. If every third filament was numbered, the font size could be larger.

We have updated Figure 1 as the review suggested. Please note that the improved Panel C (former Fig. 1C) was moved to Figure S2G.

Reviewer #3 (Remarks to the Author):

The authors uncover multiple new interesting ultrastructures of the invasion machinery of the apicomplexan parasites. This is done by cryo-FIB milling and cryo-ET of the parasite. They provide new insights into the conoid fiber structure by sub-tomogram averaging in the protruded and the retracted states and challenge the conventional spring model. They also observe for the first time multiple new “layers” of this giant complex, including a new layer of the pre-conoidal rings that they characterize by sub-tomogram averaging. The protruded and retracted states are prepared using relevant buffers, and the parasites are not actively engaged in the act of invasion. The data quality is superb and the amount of data analyzed is significant considering the challenges of in situ cryo-ET. This manuscript will open new avenues of research, including the identification of new components of this complex, and uncovering the mechanistic details of the apical complex during invasion.

We thank the reviewer for the very positive evaluation.

Major comments:

1. The panels of figure 1 and the legend of figure 1 do not match. The panels go from A to G but the legends go from A to J. Also the descriptions do not match the panels.

We sincerely apologize for all issues with the figure legends, which happened during editing and moving files between authors. We have updated the figure legend to appropriately describe the figures.

2. The quality of the tomograms is good, and it seems like most of the data is segmented. I suggest that the authors use these segmentations to present their results in 3D representations rather than 2D tomographic slices. That is true for most of the figures,

specially the main figures. The audience is interested in the structures and the relationship between different components, and that is difficult to grasp from 2D slices of busy-looking tomograms, although I appreciate the effort that the authors put in to show the kind of data that their observations are based on. I suggest that the authors find a balance between showing tomogram slices and 3D structures in the figures, mainly by reducing the number of panels with 2D slices. For example, panel F of figure 1 should be the first panel of that figure with the structures annotated on the segmentation itself. That is the most interesting result and it should be amplified.

We thank the reviewer for this comment and we agree that the 3D segmentations and visualizations are useful for readers to comprehend complex cellular organization. On the other hand, the tomographic slices are the **primary results**, whereas 3D segmentation, thresholding for isosurface renderings and the coloring is already considered **data interpretation**. Therefore, we believe that it is important to first present the primary data and then the segmented 3D renderings. However, following the reviewer's suggestion, we have tried to find a better balance in the revised figures. Specifically, we have added more examples of segmented 3D renderings (e.g. Figs. 1E, 2G and 6E), while moving some of the tomographic slices from the main figures to the supplements. In addition, we have now added cartoons of the parasite to selected figures (Figs. 1A, 2A, 5A, S7A), to orient the reader to the relevant structures. We hope the reviewer agrees that the clarity of the figures is much improved and appropriate for a broad readership.

3. I found the organization of the figures very confusing. For example, for most of the observations in the results section, there is usually more than one figure referenced, sometimes two main figures and a supplementary figure. But looking at those figures doesn't clearly get the author's observation across. The reader must do a lot of mental gymnastics to picture what is going on and has to put in a lot of effort to put the pieces together, again mainly because the figures are busy tomogram slices with annotations that are found in the legend or elsewhere. I generally found the figures to be very ineffective in getting the message across. I think each figure should have a clear message that is immediately obvious to the reader. Another example is that I was surprised that the first figure does not have a side-by-side comparison of the retracted and the protruded state using segmentations. Another example is that the different components of this giant complex are introduced on a busy tomogram slice in figure 1A, instead of a simple schematic or on figure 1F.

We thank the reviewer for the comment and we sincerely apologize for any confusions that may have been caused by mismatched figure panels and legends. The figure legends have been corrected to appropriately describe each figure.

In addition, we have introduced a cartoon overview of the parasite in Figures 1A to orient the reader, and we reuse the cartoons with selective coloring in Figures 2A, 5A, and Suppl. Fig. S7A, to highlight the relevant structure(s) covered in the respective figure. We have also re-organize our figures to illustrate our findings in the order of [APR – conoid – PCR] (meaning from the base to the apex).

As suggested by the reviewer, we have now also added a side-by-side comparison of the retracted and the protruded state to Figure 1, including a now also 3D segmented and 3D rendered retracted state (Fig. 1F).

Concerning “referencing more than one figure” when we describe structures in the results section: we understand the comment of the reviewer that this may require readers to look at multiple figures. On the other hand, we are describing many different cellular structures - often in unprecedented detail - for two different functional states of the native parasite. For clarity, the structures are described in order from base to tip, and from cytoskeleton to secretory organelles. However, most tomographic slices depict several structures. In order to not repeat images and increase the amount of figure panels, we prefer to reference back to previous figure panels where useful; we feel this is an acceptable compromise. However, we have also revised some of the figure labeling and legends to hopefully improve clarity of the figures.

4. A supplemental figure describing the genetic tree of the different species discussed in the manuscript would be helpful. Otherwise, the conclusions regarding the differences between the apical complex of different species should be made more clear.

We thank the reviewer for this suggestion. We have added a new supplementary figure S1 to the revised manuscript, showing a phylogenetic tree of apicomplexan parasites and a handful of other alveolates. We have annotated the tree with information concerning differences in the apical complex between the genera.

5. The introduction starts with a paragraph about the apical complex, it's followed by a paragraph about the genetic tree and the micronemes and rhoptries, and then the next paragraph goes back to discussing the apical complex. I think the flow would be better if the third paragraph switched places with the second paragraph.

We agree with the reviewer that swapping these paragraphs improves the flow of the introduction and have made that change in the manuscript.

6. The paragraph describing in situ cryo-ET that starts at line 65 is lengthy and I'm not sure that there is a need for such a long introduction to the technique. I also found the bottom-up and top-down terminology unnecessary and unclear. Maybe it would suffice to say that in situ cryo-ET provides context to the structural analysis and that it might reveal structures that were not seen before in reconstituted studies.

We thank the reviewer for the comment. We shortened the paragraph and removed the “bottom-up/top-down” terminology, by removing the first three sentences, so that the paragraph in the revised manuscript now starts with “Cellular cryo-ET is a powerful imaging technique ...”.

7. *The choice of Toxoplasma gondii and Neospora caninum for the in situ cryo-ET analysis is not clear (Line 77). why these two specifically?*

We chose these organisms because a) *Toxoplasma* is arguably the most successful parasite in the world, and b) because *N. caninum* is a very closely related organism, yet it is BSL1 (rather than BSL2 like *Toxoplasma*), which reduces issues with handling rapidly frozen – and thus considered “native” - samples in electron microscopes (FIB and TEM). We clarify both points in the manuscript on page 4: “*We have applied cryo-ET to the coccidian apicomplexan parasites Toxoplasma gondii, arguably the most widespread and successful parasite in the world*”, and on page 6: “*To generate these samples, we used the NC1 strain of Neospora caninum, which is a BSL1 organism closely related to the human pathogen Toxoplasma gondii.*” Also, we list in the methods which figures depict *Toxoplasma* or *Neospora* , respectively: “*Detergent-extracted Toxoplasma cells and the associated subtomogram averages are displayed in Figure 7G and Supplemental Figures S2A-F,I-K, S4A-B, and S6I. Tomographic reconstructions of cryo-FIB-milled N. caninum cells, and the corresponding subtomogram averages and data analyses are presented in Figures 1-6, 7A-F,H-K and Supplemental Figures S2G,H, S3, S4C-L, S5, S6A-H, and S7-S8.*”.

8. *Line 80: “without the compression and deformation that plague unmilled samples”. This phrasing is misleading because not all unmilled samples are compressed.*

We have revised the sentence as follows to clarify its meaning: “*without the compression and deformation artifacts that have plagued studies of unmilled intact apicomplexan samples*”. We cite all previously published cryo-ET studies of apicomplexan parasites - all of which did not utilize cryo-FIB milling and imaged/visualized compressed samples. [Please note that some of these reports do not specifically mention the compression in their text, but the figures and especially the movies of 3D renderings/segmentations unambiguously show the sample compression].

9. *The title of the first result “Cryo-FIB-milling of intact parasites...” is very generic and sounds like what one would find in a methods paper. I think the first result should be a clear statement about the observed structures, and not a description of in situ cryo-ET. The content of this section is also focused on methods. I think it’s clear to the community what each of these methods provide.*

We somewhat disagree with the reviewer on this point, because a) the broad readership of Nature Communications might not (yet) be familiar with the technical advances/advantages of cryo-FIB milling, and b) application of cryo-FIB milling in our study was critical for being able to reconstruct uncompressed samples – which is most likely the reason why our results differ significantly from previous reports, supporting a completely different translocation mechanism of the apical complex than previously proposed. Therefore, we prefer to keep this somewhat technical, yet important section. However, we have revised

the title of the section to “*Cryo-FIB-milling followed by cryo-electron tomography preserves the 3D structure of the parasite cytoskeleton.*”

10. Line 114: 10 micro molar?

We thank the reviewer for pointing out this typo, which has been corrected in the revised manuscript “*10 μ M*”.

11. Line 204: *the reasoning sounds circular. It sounds like the authors are just describing what they mean by conformational change in the context of the retracted and protruded states. I think the opening sentence should state that the authors set out to investigate the spring model by looking at gross changes in the dimensions.*

As per the reviewer's suggestion, we now begin the paragraph in question with the sentence: “*We therefore sought to assess whether the conoid movements behaved according to the spring-like model involving deformation of the conoid and conoid fibers*”

12. *The conclusion that the spring model is incorrect and the introduction of the rigid body model are not corroborated enough. These states were induced chemically, and the organism is not actively invading a host, so one could argue that the protruded state might be an intermediate/alternative state and that's why the spring-like motion is absent. The authors should rule out these possibilities in clear statements. The authors rather argue that in situ cryo-ET provides close-to-native conditions (which is true) therefore, the induced state is exactly the protruded state.*

There appears to be some confusion as to the need/relevance of having host cells present to induce conoid protrusion and/or microneme secretion in extracellular parasites. The lack of host cells in our sample is immaterial, because microneme secretion (e.g. of adhesins) is absolutely required for initiating movement from the parasite vacuole and for moving to locate a new host cell (before invasion or contact with host cells). Similarly, the current understanding is that addition of ionophore does not block conoid movement, but changes the probability of finding the conoid in the protruded versus retracted state. In fact, addition of ionophore to parasites (i.e. increasing their intracellular calcium) that are inside host cells induces egress, which requires multiple complex steps (i.e. secretion of micronemes, release of the “exit” factors to lyse the host cell and induction of motility). We have clarified this in the revised manuscript by adding the following sentence: “*The conoid of extracellular parasites protrudes and retracts continuously - with and without host cells present, but just before plunge-freezing, the parasites were either prepared in an intracellular-like buffer³⁷ or incubated with 10 μ M calcium ionophore for 10 min. so that the majority of parasites have conoids preferably in the retracted or protruded states⁶, respectively, without blocking conoid motility or cellular functions like secretion.*”

Nevertheless, the reviewer is correct that we have no data regarding the apical complex structure in the context of an actively invading parasite, and it is entirely possible that there may be changes in conoid movement or structure in this context. Therefore, we have

added the following sentence to the last paragraph of the discussion: “*The next frontier in understanding the mechanics of the apical complex will be in capturing parasites during host cell invasion. It is possible that components of the apical complex will undergo some conformational change when the parasite is in intimate contact with a host cell membrane.*”

13. Line 240: *at 3 nm resolution it is a big statement to say that positioning of the protofilaments is indistinguishable between the two states. Figure 5E shows differences between the two states and the docking in figure 5C and 5D does not seem to be tight.*

The reviewer is, of course, correct. The data are not high enough resolution to obtain a “tight” docking of the protofilaments into the density. For this reason, we have tried to make clear that within the error of our data and the fit the protofilaments are indistinguishable. We have updated the text to read “*each of the 9 protofilament subunits appear positioned indistinguishably between the two states within the resolution of the data and error of the fit.*” Please note that **previous studies with similar resolution** of their subtomogram averages of conoid fibers – but of compressed conoids – did observe differences at this resolution; therefore, our observation is still highly relevant and important - despite of the moderate resolution (when compared to higher resolution single-particle cryo-EM or cryo-ET of thinner samples).

14. Line 257: *The increase in resolution is not interpretable because merely the higher number of particles contributing to the map could inflate the estimated resolution. It certainly cannot be used to prove the rigidity of the structure, specially since it’s not clear how much difference is a meaningful difference in this context. I think this data serves as a lack of evidence for severe compression or expansion, but It does not prove that the two states have identical structures.*

We agree with the reviewer and apologize for the confusion on this point. We did not mean to argue at this point that the modest increase in resolution of the global average was evidence for the rigidity of the structure. The goal at this point was to generate a global average with higher signal-to-noise ratio for further analysis. We have modified the text at this point to read: “*The resulting average of the two states shows markedly improved signal-to-noise ratio and a resolution of ~2.8 nm at the 0.5 FSC criterion (or ~2.2 nm at the 0.143 FSC criterion; Supplemental Figure S6B) and reveals density ...*”

15. *There is a lack of referencing figures in the discussion.*

We have followed what we considered the standard convention to not repeat references to data figures in the Discussion. If this is against Nat. Comm. editorial policy, we are, of course, happy to alter the manuscript accordingly.

16. *The authors milled 178 lamella, collected 167 tomograms, of which 20 had the apical complex. The authors should comment on the nature of the uncertainties that limited the throughput during each step (milling and data collection) in the methods section.*

We thank the reviewer for the suggestion and have added the following content in the methods section: *“There are several factors that contribute to the low yield of the apical complex in our experiments. Firstly, we have plunge-frozen the intact parasites in a relatively thick (>1 μm) layer of ice. It is challenging to determine the apical or basal ends of the ice-embedded cell in the cryo-FIB milling instrument. Secondly, during the cryo-FIB milling step, the ice thickness was reduced from more than 1 μm to 150-200 nm, removing more than 80% of the volume. Thus, the probability of placing the lamella exactly in the region of the about 300 nm wide conoid is relatively low compared to accidentally milling the apical complex away during the thinning step. Finally, about 10~15% lamellae are damaged or surface-contaminated during the transfer step from the milling instrument to the TEM. Future application of fluorescence-guided cryo-FIB milling and autoloader systems for more direct transfer of FIB milled samples into the TEM could address these issues and increase experiment throughput.”*

17. *I think there are two story lines in this paper that do not necessarily go well together. One is focused on methods and comparing methods in detail. And one is focused on structure and mechanism. I think the flow would be much better and the manuscript would be much easier to read and the results would be more impactful if the authors stick to one story line, that is the structural analysis.*

As mentioned above, we somewhat disagree with the reviewer on this point, because the technical comparison sets the stage for why our samples are better preserved compared to previous studies. We believe this is a critical line of evidence that supports our (paradigm-shifting) observations and interpretations. However, we have re-organized the figures and main texts to increase emphasis on our structural findings and reduce the technical analysis.

REVIEWERS' COMMENTS

Reviewer #1 (Remarks to the Author):

I am satisfied with the changes the authors incorporated in the revised manuscript. Overall, this dramatically improves the clarity, and thus the impact, of the exciting results presented. At this point, I recommend that the manuscript is accepted for publication.

Reviewer #2 (Remarks to the Author):

I am happy with the changes to the manuscript made by the authors in response to my first review. In my opinion, it is now acceptable for publication.

Reviewer #3 (Remarks to the Author):

The authors have done an excellent job of addressing all the comments and concerns, and the manuscript is in a good shape for publication.